# Genome-wide association study of actinic keratosis identifies new susceptibility loci implicated in pigmentation and immune regulation pathways

Yuhree Kim [1,2], Jie Yin[3], Hailiang Huang[4,5], Eric Jorgenson [6], Hélène Choquet [3✉] & Maryam M. Asgari[1,2✉]

Actinic keratosis (AK) is a common precancerous cutaneous neoplasm that arises on chronically sun-exposed skin. AK susceptibility has a moderate genetic component, and although a few susceptibility loci have been identified, including *IRF4*, *TYR*, and *MC1R*, additional loci have yet to be discovered. We conducted a genome-wide association study of AK in non-Hispanic white participants of the Genetic Epidemiology Research on Adult Health and Aging (GERA) cohort ($n = 63,110$, discovery cohort), with validation in the Mass-General Brigham (MGB) Biobank cohort ($n = 29,130$). We identified eleven loci ($P < 5 \times 10^{-8}$), including seven novel loci, of which four novel loci were validated. In a meta-analysis (GERA + MGB), one additional novel locus, *TRPS1*, was identified. Genes within the identified loci are implicated in pigmentation (*SLC45A2, IRF4, BNC2, TYR, DEF8, RALY, HERC2,* and *TRPS1*), immune regulation (*FOXP1* and *HLA-DQA1*), and cell signaling and tissue remodeling (*MMP24*) pathways. Our findings provide novel insight into the genetics and pathogenesis of AK susceptibility.

[1] Department of Dermatology, Massachusetts General Hospital, Boston, MA, USA. [2] Department of Population Medicine, Harvard Medical School and Harvard Pilgrim Health Care Institute, Boston, MA, USA. [3] Division of Research, Kaiser Permanente Northern California, Oakland, CA, USA. [4] Analytic and Translational Genetics Unit, Massachusetts General Hospital, Boston, MA, USA. [5] Department of Medicine, Harvard Medical School, Boston, MA, USA. [6] Regeneron Genetics Center, Tarrytown, NY, USA. ✉email: Helene.Choquet@kp.org; pores@mgh.harvard.edu

Actinic keratoses (AKs) are keratinocyte-derived neoplasms that arise on skin exposed to chronic ultraviolet (UV) radiation[1]. They are highly prevalent among older individuals with light pigmentation, with prevalence estimates ranging from 11 to 60% in non-Hispanic whites (NHW) over 40 years of age[2,3]. Importantly, AKs can progress to develop into keratinocyte carcinoma (KC), particularly cutaneous squamous cell carcinoma (cSCC), which is among the most common and costly malignancies among NHWs[4–6]. The underlying pathogenesis of AK includes alterations in the pathways regulating cell growth and differentiation, inflammation, and immunosuppression caused by UV radiation, tissue remodeling, oxidative stress, and impaired apoptosis[7].

Characterizing the genetic factors influencing AK susceptibility is an essential step toward understanding the pathogenesis of keratinocyte neoplasia. Genetic susceptibility to AK has been implicated by a genome-wide association study (GWAS) performed in a European cohort, which identified three pigmentation-related loci (i.e., IRF4, TYR, and MC1R), explaining 2.6% of the variance in the risk of AK[8]. A subsequently published genome-wide compound heterozygote scan reported 15 AK-associated loci, three of which (KCNK5/KCNK17, PAQR8/GSTA2, and KCNQ5/KHDC1) were replicated that were not related to pigmentation pathways[9]. AK has a moderate genetic component with an array-heritability estimate of ~17.0%[8]. However, many of the reported genetic risk loci have not been validated, and the genetic etiology of AK remains largely unknown.

To address these knowledge gaps, we performed a GWAS of AK among 63,110 non-Hispanic white individuals (16,352 AK cases and 46,758 controls) in the Kaiser Permanente Genetic Epidemiology Research on Adult Health and Aging (GERA) cohort (discovery cohort). We then validated our findings in an independent cohort of 29,130 NHW individuals in the Mass-General Brigham (MGB) Biobank cohort (5110 AK cases and 24,020 controls). Our findings validate the previously reported susceptibility loci and identify novel loci with functional roles in pigmentation and immune system pathways, highlighting biological pathways through which human genetic variation impacts keratinocyte carcinogenesis.

## Results

**GERA cohort.** We conducted the primary discovery analysis of AK in 16,352 AK cases and 46,758 controls from the NHW GERA sample. As compared to controls (subjects without AK diagnosis), subjects with AK were more likely to be older (65.6 vs. 58.7 years) and male (49.6% vs. 38.2%)(Table 1), consistent with previous studies[2,10].

We identified eleven genome-wide significant ($P < 5 \times 10^{-8}$) loci associated with AK (Table 2 and Fig. 1), of which seven loci have not been previously reported. These included FOXP1 (lead SNP rs62247035), SLC45A2 (lead SNP rs16891982), HLA-DQA1 (lead SNP rs9271377), BNC2 (lead SNP rs12350739), RALY (lead

SNP rs6059655), and MMP24 (lead SNP rs2425025). We confirmed genome-wide association with AK at the previously reported IRF4, TYR, and MC1R loci[8]. The QQ plot is presented in Fig. 2a. SNPs rs6059655 in RALY and rs2425025 in MMP24 on 20q11 are in linkage disequilibrium ($r^2 = 0.85$).

**Conditional analyses identified an additional locus.** A conditional analysis in the GERA cohort identified one additional SNP (rs35063026), which resides in the SPATA33 gene within the previously reported MC1R/DEF8 locus (OR 1.43, $P = 3.7 \times 10^{-50}$). This SNP was not a proxy variant of the previously reported SNP rs1805008. These two SNPs (rs35063026 and rs1805008) are ~0.25 Mb apart and in linkage equilibrium ($r^2 = 0.005$), suggesting that they are independent signals in the same locus. We generated a regional association plot at the MC1R/DEF8 locus to illustrate the multiple independent signals at this genomic region (Supplementary Fig. 1).

**Sensitivity analysis.** To examine whether identified AK-associated loci were driven by cSCC risk, we performed a sensitivity analysis limiting the cohort to participants in GERA without previously diagnosed cSCC. Eight out of the eleven AK-associated loci (SLC45A2, IRF4, BNC2, HERC2, DEF8, SPATA33, RALY, and MMP24) were confirmed to be associated with AK at a genome-wide level of significance ($P < 5 \times 10^{-8}$). The remaining three AK-associated loci (FOXP1, HLA-DQA1, and TYR) reached Bonferroni-level of significance ($P = 4.5 \times 10^{-3} = 0.05/11$; 11 SNPs tested) (Supplementary Table 1). The direction of the effect of the risk allele of AK-susceptibility loci was consistent with those of the full study cohort.

**Validation in the MGB Biobank cohort.** The validation cohort consisted of 5110 AK cases and 24,020 controls from the MGB Biobank cohort. Similar to our findings in the GERA cohort, MGB Biobank AK cases were more likely to be older (72.3 vs. 58.7 years) and male (53.7% vs. 46.1%) when compared to controls (Table 1). We assessed the 11 lead SNPs identified in the GERA cohort for validation in the MGB Biobank cohort. We found five SNPs validated at a genome-wide significant level, including rs16891982 at SLC45A2, rs12203592 at IRF4, rs1126809 at TYR, rs4268748 at DEF8, and rs35063026 at SPATA33 (Table 2). Three additional SNPs were validated at Bonferroni significance ($P < 4.5 \times 10^{-3} = 0.05/11$; 11 SNPs tested) including rs12350739 in BNC2, rs6059655 in RALY, and rs2425025 in MMP24. The remaining loci (rs62247035 in FOXP1, rs9271377 in HLA-DQA1, rs12916300 in HERC2) did not reach statistical significance, although their direction of effect was consistent with those of the discovery cohort.

**Confirmation of previously reported AK-associated loci.** We investigated the three previously reported AK-associated SNPs in

**Table 1 Characteristics of GERA and MGB Biobank NHW subjects with AK (cases) and controls.**

| | | GERA | | | MGB Biobank | | |
|---|---|---|---|---|---|---|---|
| | | N (%) | AK cases | Controls | N (%) | AK cases | Controls |
| All | | 63,110 | 16,352 (25.9) | 46,758 (74.1) | 29,130 | 5110 (17.5) | 24,020 (82.5) |
| Age[a] (years) mean ± SD | | 60.5 ± 29.1 | 65.6 ± 38.9 | 58.7 ± 24.5 | 61.1 ± 16.6 | 72.3 ± 10.5 | 58.7 ± 16.6 |
| Sex | Female | 37,124 (58.8) | 8241 (50.4) | 28,883 (61.8) | 15,319 (52.6) | 2367 (46.3) | 12,952 (53.9) |
| | Male | 25,986 (41.2) | 8111 (49.6) | 17,875 (38.2) | 13,811 (47.4) | 2743 (53.7) | 11,068 (46.1) |

SD standard deviation.
[a]Age at entry of the cohort.

**Table 2 Lead genome-wide significant SNP for each independent locus identified in the GERA discovery cohort (*n* = 63,110) and validation in MGB Biobank cohort (*n* = 29,130).**

| SNP | Chr | Pos | Locus | Gene[a] | MAF | Ref/Eff allele | GERA discovery cohort | | MGB Biobank validation cohort | |
|---|---|---|---|---|---|---|---|---|---|---|
| | | | | | | | OR (95% CI) | *P*-value | OR (95% CI) | *P*-value |
| rs62247035 | 3 | 71,546,358 | 3p13 | FOXP1 | 0.34 | G/A | 1.1 (1.07, 1.13) | 1.1E-11 | 1.05 (1.00, 1.10) | 5.1E-02 |
| rs16891982 | 5 | 33,951,693 | 5p13 | SLC45A2 | 0.04 | G/C | 0.48 (0.40, 0.56) | 8.4E-71 | 0.63 (0.55, 0.71) | 5.6E-13 |
| rs12203592 | 6 | 396,321 | 6p25 | IRF4 | 0.17 | C/T | 1.57 (1.54, 1.60) | 2.0E-155 | 1.40 (1.32, 1.48) | 8.2E-33 |
| rs9271377 | 6 | 32,587,165 | 6p21 | HLA-DQA1 | 0.38 | T/G | 1.09 (1.06, 1.12) | 2.5E-10 | 1.05 (1.01,1.11) | 3.1E-02 |
| rs12350739 | 9 | 16,885,017 | 9p22 | BNC2 | 0.43 | A/G | 0.88 (0.85, 0.90) | 2.0E-22 | 0.90 (0.86, 0.95) | 3.9E-05 |
| rs1126809 | 11 | 89,017,961 | 11q14 | TYR | 0.28 | G/A | 1.13 (1.10, 1.16) | 1.2E-17 | 1.16 (1.10, 1.22) | 9.3E-09 |
| rs12916300 | 15 | 28,410,491 | 15q13 | HERC2 | 0.26 | T/C | 0.88 (0.85, 0.91) | 5.3E-15 | 0.93 (0.88, 0.98) | 6.0E-03 |
| rs4268748 | 16 | 90,026,512 | 16q24 | DEF8 | 0.26 | T/C | 1.29 (1.26, 1.32) | 9.8E-56 | 1.17 (1.11, 1.23) | 2.2E-09 |
| rs35063026[b] | 16 | 89,736,157 | 16q24 | SPATA33 | 0.08 | C/T | 1.43 (1.38, 1.48) | 3.7E-50 | 1.32 (1.22, 1.44) | 6.6E-11 |
| rs6059655 | 20 | 32,665,748 | 20q11 | RALY | 0.08 | G/A | 1.33 (1.28, 1.38) | 8.8E-34 | 1.20 (1.11, 1.31) | 8.0E-06 |
| rs2425025 | 20 | 33,847,154 | 20q11 | MMP24 | 0.06 | A/G | 1.31 (1.25, 1.39) | 1.5E-23 | 1.16 (1.07, 1.27) | 6.5E-04 |

*SNP* single-nucleotide polymorphism, *Chr* chromosome, *Pos* position, *MAF* minor allele frequency, *Ref/Eff allele* Reference/effect allele, *OR* odds ratio, *CI* confidence interval.
[a]Novel loci are underlined.
[b]Independently AK-associated SNP on conditional analysis.

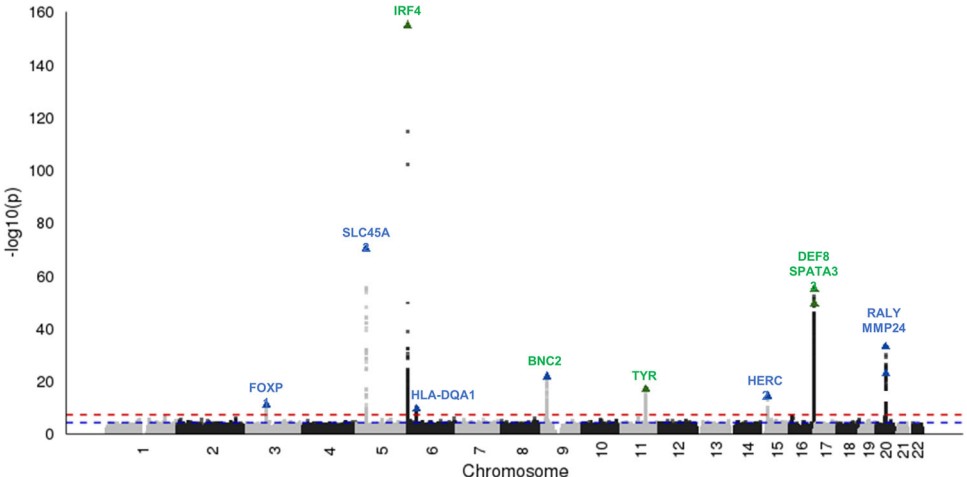

**Fig. 1 Manhattan plot of GWAS of actinic keratosis of GERA discovery cohort (*n* = 63,110).** The y axis represents log-scale *P*-values, and loci with the smallest *P* obtained from the logistic regression are labeled with the nearest or corresponding gene's name. Known loci are in green triangles and newly identified loci are in blue triangles.

both the GERA and MGB Biobank cohorts (Table 3). All 3 SNPs were replicated at a genome-wide level of significance in both cohorts. Of note, the most significant SNP in the 16q24 locus in the current study was rs4268748 in the *DEF8* gene, whereas in the previous study, it was rs139810560 in the *MC1R* gene. These genes are ~14.8 kb apart and in low linkage disequilibrium ($r^2 = 0.18$), suggesting that this region may be subject to adaptive selection[11–13].

**A meta-analysis of GERA and MGB Biobank.** We conducted a meta-analysis combining the GERA and MGB Biobank cohorts to increase the power to detect additional novel loci (QQ plot in Fig. 2b). In addition to the 12 independent loci identified in the GERA discovery or conditional analysis, we identified one locus, *TRPS1* (lead SNP rs7832568, OR = 1.07, $P = 2.84 \times 10^{-8}$), that was not previously reported (Table 4 and Fig. 3). Regional association plots at the novel AK-susceptibility loci are presented in Supplementary Fig. 2a–g.

**Gene and pathway prioritization.** We conducted gene-based and biological pathway prioritization analyses using Versatile Gene-

based Association Study 2 (VEGAS2) software implemented in a command-line tool (https://vegas2.qimrberghofer.edu.au). VEGAS2 integrative tool aggregates association strength of individual markers into pre-specified biological pathways[14,15]. Using a mapping threshold of 10 kb upstream and downstream of gene boundaries, we found 55 genes that reached a significant threshold after correcting for multiple testing of 23,051 genes ($P < 2.17 \times 10^{-6}$) (Supplementary Table 2). We found a significant association with AK for the *ANXA9* gene on chromosome 1, which was not identified in the discovery GWAS or meta-analysis. The pathway-based analysis identified five pathways/gene-sets that were significantly enriched after correcting for multiple testing of 9,736 pathways/gene-sets ($P < 5.14 \times 10^{-6}$) (Supplementary Table 3). These included melanin or secondary-metabolite biosynthesis pathways. Significant results (Benjamini–Hochberg false discovery rate [FDR] control approach with FDR of 0.1) of gene-based analysis and pathways/gene-sets analysis are presented in Supplementary Tables 2 and 3.

**Heritability estimate for AK.** We estimated SNP-based heritability in the GERA NHW sample using a linkage disequilibrium

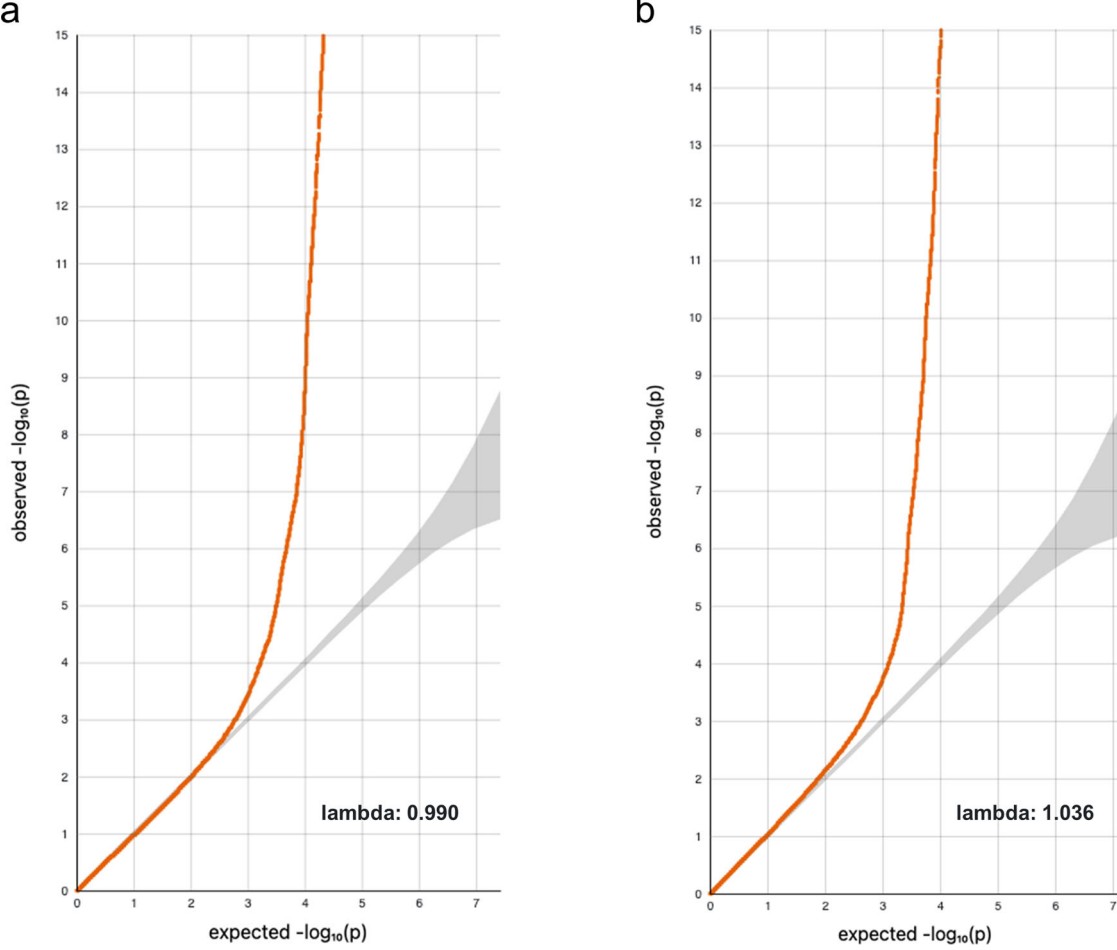

**Fig. 2 QQ plots of the GWAS. a** GWAS of GERA discovery cohort ($n = 63,110$). **b** GWAS of meta-analysis combining GERA and MGB Biobank cohorts ($n = 92,240$). Lambda indicates inflation factor.

**Table 3 Replication of previous AK GWAS results in the GERA discovery cohort and MGB Biobank cohort.**

| SNP | Chr | Pos | Locus | Gene | Ref/Eff allele | GERA discovery cohort | | MGB Biobank cohort | |
|-----|-----|-----|-------|------|----------------|-----------------------|---------|--------------------|---------|
| | | | | | | OR (95% CI) | *P*-value | OR (95% CI) | *P*-value |
| rs12203592 | 6 | 396,321 | 6p25 | IRF4 | C/T | 1.57 (1.54, 1.60) | 2.0E-155 | 1.40 (1.32, 1.48) | 8.2E-33 |
| rs1393350 | 11 | 89,011,046 | 11q14 | TYR | G/A | 1.12 (1.09, 1.16) | 9.9E-15 | 1.16 (1.10, 1.22) | 1.3E-08 |
| rs139810560 | 16 | 90,011,739 | 16q24 | MC1R | C/A | 1.30 (1.24, 1.37) | 6.4E-27 | 1.19 (1.09, 1.29) | 9.6E-05 |

score regression (LDSC), and we found an SNP-based heritability estimate of 0.077 ($h^2_{SNP}$; 95% CI 0.05–0.10).

## Discussion

Our large GWAS of AK confirmed the previously reported association between SNPs in genes related to the pigmentation pathway (*IRF4* on chromosome 6p25, *TYR* on chromosome 11q14, and *MC1R* on chromosome 16q24)[8,9], and also has identified previously unreported AK-associated SNPs at novel loci (*FOXP1* on 3p13, *SLC45A2* on 5p13, *HLA-DQA1* on 6p21, *TRPS1* on 8q23, *BNC2* on 9p22, *HERC2* on 15q13, and *RALY* and *MMP24* on 20q11). These novel loci harbor genes implicated in pigmentation, immune regulation, and cell signaling. Several loci are in or near genes in the pigmentation pathways that are likely to be relevant to AK and associated with skin tanning ability and KC risk[16–19]. *IRF4* activates the melanogenic enzyme *TYR* expression, while *RALY-ASIP* antagonizes the pathway. *SLC45A2* and *HERC2/OCA2* regulate melanin production, and *BNC2* may

regulate the expression of pigmentation genes[20]. Further, we identified AK-associated pathways, including the melanin synthesis process. Melanin pigment molecules form a coat around the nucleus of epidermal keratinocytes that may protect the keratinocytes from UV-induced DNA damage, which can lead to AK development.

The predominance of AK susceptibility loci associated with pigmentation genes suggests the critical role of components of the pigmentation pathway for AK risk. Also, it confirms the well-known heritable risk factors of AK, such as hair and eye colors or fair skin[5]. Previous GWAS of the Rotterdam population reported that *IRF4*, *MC1R*, and *TYR* genes might increase AK risk by affecting pigmentation and oncogenic functions[8]. A subsequent compound heterozygote scan on candidate pigmentation genes reported a suggestive association with *HERC2* loci ($P = 5.5 \times 10^{-6}$), while the association with *SLC45A2* and *BNC2* did not reach statistical significance[9]. Our strongest signal was SNP rs12203592 in *IRF4* ($P = 1.97 \times 10^{-155}$), consistent with the

**Table 4 Lead genome-wide significant SNP for each independent locus identified in the meta-analysis combining GERA and MGB Biobank cohorts (n = 92,240).**

| SNP | Chr | Pos | Locus | Gene[a] | Ref/Eff allele | Q | Dir | PR | Meta-analysis | | GERA discovery cohort | | MGB Biobank validation cohort | |
|---|---|---|---|---|---|---|---|---|---|---|---|---|---|---|
| | | | | | | | | | OR (95% CI) | P-value | OR (95% CI) | P-value | OR (95% CI) | P-value |
| rs7638354 | 3 | 71,548,328 | 3p13 | FOXP1 | T/A | 0.13 | ++ | 1.7E-04 | 1.09 (1.06, 1.11) | 4.4E-12 | 1.1 (1.07, 1.13) | 1.4E-11 | 1.05 (1.00, 1.11) | 3.6E-02 |
| rs16891982 | 5 | 33,951,693 | 5p13 | SLC45A2 | G/C | 0.00 | –– | 4.9E-06 | 0.52 (0.48, 0.55) | 8.1E-80 | 0.48 (0.40, 0.56) | 8.4E-71 | 0.63 (0.55, 0.71) | 5.6E-13 |
| rs4455710 | 6 | 32,608,858 | 6p21 | HLA-DQA1 | C/T | 0.41 | ++ | 1.6E-11 | 1.08 (1.06, 1.11) | 1.6E-11 | 1.09 (1.06, 1.11) | 3.8E-10 | 1.06 (1.02, 1.12) | 1.0E-02 |
| rs12203592 | 6 | 396,321 | 6p25 | IRF4 | C/T | 0.00 | ++ | 5.7E-12 | 1.52 (1.48, 1.56) | 7.8E-184 | 1.57 (1.54, 1.60) | 2.0E-155 | 1.4 (1.32, 1.48) | 8.2E-33 |
| rs7832568 | 8 | 116,498,798 | 8q23 | TRPS1 | A/C | 0.40 | ++ | 2.8E-08 | 1.07 (1.04, 1.09) | 2.8E-08 | 1.06 (1.03, 1.09) | 1.2E-05 | 1.09 (1.04, 1.14) | 4.7E-04 |
| rs12350739 | 9 | 16,885,017 | 9p22 | BNC2 | A/G | 0.24 | –– | 6.0E-16 | 0.88 (0.86, 0.90) | 5.6E-26 | 0.89 (0.86, 0.91) | 5.1E-19 | 0.9 (0.85, 0.94) | 7.0E-06 |
| rs1126809 | 11 | 89,017,961 | 11q14 | TYR | G/A | 0.46 | ++ | 8.6E-25 | 1.14 (1.11, 1.17) | 8.6E-25 | 1.13 (1.10, 1.16) | 1.2E-17 | 1.16 (1.10, 1.22) | 9.3E-09 |
| rs12916300 | 15 | 28,410,491 | 15q13 | HERC2 | T/C | 0.13 | –– | 6.1E-06 | 0.89 (0.87, 0.92) | 3.4E-16 | 0.88 (0.85, 0.91) | 5.3E-15 | 0.93 (0.88, 0.98) | 6.0E-03 |
| rs4268748 | 16 | 90,026,512 | 16q24 | DEF8 | T/C | 0.00 | ++ | 2.0E-05 | 1.25 (1.22, 1.28) | 2.3E-61 | 1.29 (1.26, 1.32) | 9.8E-56 | 1.17 (1.11, 1.23) | 2.2E-09 |
| rs6059655 | 20 | 32,665,748 | 20q11 | RALY | G/A | 0.04 | ++ | 1.3E-06 | 1.30 (1.25, 1.35) | 3.1E-37 | 1.33 (1.28, 1.38) | 8.8E-34 | 1.20 (1.11, 1.31) | 8.0E-06 |
| rs2425025 | 20 | 33,847,154 | 20q11 | MMP24 | A/G | 0.02 | ++ | 3.1E-04 | 1.27 (1.21, 1.33) | 6.6E-25 | 1.31 (1.25, 1.39) | 1.5E-23 | 1.16 (1.07, 1.27) | 6.5E-04 |

SNP single-nucleotide polymorphism, Chr chromosome, Pos position, MAF minor allele frequency, Ref/Eff allele reference/effect allele, OR odds ratio, CI confidence interval.
[a]Novel loci are underlined.
[b]Independently AK-associated SNP on conditional analysis.

previous study. SNP rs12203592 lies within an intronic regulatory region of the interferon regulatory factor 4 (*IRF4*) gene that impacts skin pigmentation by modulating enhancer-mediated transcriptional regulation and physically interacting with the *IRF4* gene promoter in an allele-specific manner[21]. This SNP has been associated with pigmentation, hair color, eye color, freckles, skin sensitivity to sun exposure, and skin cancers, including cSCC, basal cell carcinoma (BCC), and melanoma[18,22–28]. *IRF4* cooperates with the melanocyte master regulator, microphthalmia-associated transcription factor (MITF), to activate the tyrosinase (TYR) expression that catalyzes melanin production and other pigments from tyrosine by oxidation[29,30]. In the previously published GWAS[8], AK-susceptibility SNP in *TYR* locus (rs1393550) did not reach a genome-wide level of significance, likely due to limited sample size. The association of rs1393350 and AK risk reached a genome-wide level of significance in our study, and we identified an additional lead SNP rs1126809 at the *TYR* locus. SNP rs1126809 has been associated with a low tan response and increased risk of keratinocyte carcinomas and melanoma[16,18,24,25,31], and it may cause changes at the post-translational modification site, leading to dysregulation of melanin synthesis within the melanosomes[32]. Similarly, SNP rs16891982 lies in *SLC45A2*, which encodes a transporter protein that mediates melanin synthesis, correlates with reduced melanin content in cultured human melanocytes[33]. The variant has been associated with pigmentation and melanoma[23,31]. In addition, SNP rs12350739, an intergenic SNP of basonuclin 2 (*BNC2*), and the highly conserved surrounding region function as enhancers regulating *BNC2* transcription in human melanocytes[34]. Variants in the *BNC2* locus have been associated with skin color, freckling, and KC[16,24,25,34,35]. SNP rs12916300 is an intronic variant in *HERC2* (HECT and RLD domain containing E3 ubiquitin-protein ligase 2). Individual SNPs at this locus and the nearby gene *OCA2* have been associated with pigmentation variability as well as cSCC risk[24,36].

Interestingly, most AK-associated loci have been previously reported as cSCC-associated loci suggesting common biological pathways in keratinocyte carcinogenesis[24]. While cSCC can arise either *de novo* or from a preexisting AK, the annual risk of cSCC for subjects with multiple AKs ranged from 0.15 to 80%[37]. A previous study on gene expression patterns demonstrated that AK and cSCC were genetically related[38]. The direction of the effect of the risk allele of AK-susceptibility loci is similar to what was found in the previous GWAS of cSCC risk[24,25]. In addition to those loci related to the pigmentation pathway, variants in *HLA-DQA1* were associated with AK risk. Class II HLA genes encode major histocompatibility complex (MHC) molecules that bind antigenic peptides presented by antigen-presenting cells and deliver them to the T-cell receptors on T cells. Increased AK incidence among immunosuppressed subjects may suggest a role for HLA antigens and immune response in AK pathogenesis[39–42]. Interestingly, rs4455710 at *HLA-DQA1* locus that was identified in our meta-analysis is the same lead SNP that was identified as a cSCC risk locus in the previous GWAS[24]. We identified rs9271377 at the *HLA-DQA1* locus in our discovery cohort, and this SNP was in strong linkage disequilibrium with rs4455710 ($r^2 = 0.6483$). This may suggest shared immune effects involving AK and SCC, given that AKs can be a precursor to cSCC among individuals with both AKs and SCC. In addition, SNP rs62247035 at 3p13 is intronic in forkhead box P1 (*FOXP1*), a gene that encodes a transcriptional factor that regulates lymphocyte development and whose abnormal expression has been demonstrated in various human cancers. *FOXP1* has been reported as a negative regulator of anti-tumor immune responses via its regulation on chemokine expression and MHC class II expression[43,44]. Notably, common variants in *FOXP1* have been

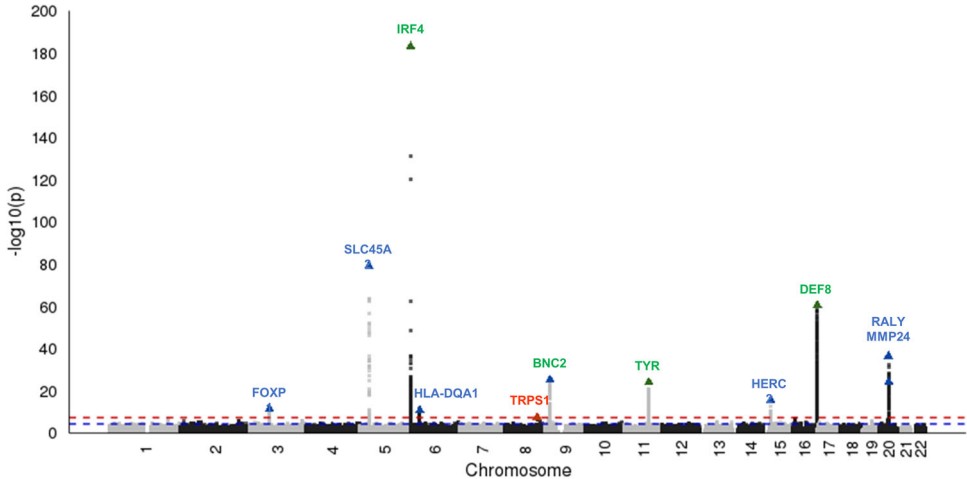

**Fig. 3 Manhattan plot of GWAS meta-analysis of actinic keratosis combining results of GERA and MGB Biobank cohorts ($n = 92,240$).** The y axis represents log-scale P-values, and loci with the smallest P obtained from the logistic regression are labeled with the nearest or corresponding gene's name. Known loci are in green triangles, newly identified loci in discovery analysis are in blue triangles, and newly identified loci in meta-analysis are in orange triangles.

associated with cSCC and BCC in previous GWAS[16,18,24]. The SNP rs62246017 in *FOXP1* was previously associated with cSCC, and it was in linkage disequilibrium with rs7638354 identified in the current study ($r^2 = 0.882$). Interestingly, recent studies showed that immunotherapy of AK reduced the risk of SCC development by inducing T-cell immunity[45,46]. The finding of shared immune-related genomic loci associated with both AK and cSCC risk suggests that immune-related pathways may hold promise for novel therapeutic options in keratinocyte carcinogenesis.

Multiple SNPs were identified in 16q24 (*DEF8* and *SPATA33* locus) that are associated with pigmentation traits, tanning response, and skin cancer risk[16,18,22–25,28,47]. While the previously published GWAS identified rs139810560 in *MC1R* at 16q24 to be associated with AK[8], and rs1805007 in *MC1R* was the most significant SNP at this locus in our validation cohort. Common DNA variants at the *MC1R* gene encoding melanocortin one receptor on the melanocytes that produce a melanin pigment have been associated with tanning ability, hair color, and KC and melanoma risk[16,18,23,25,28,31]. Previous studies found multiple SNPs independently related to hair color near the *MC1R* locus[23,48]. We found rs4268748 in *DEF8* at the 16q24 locus to be most significantly associated with AK risk and rs35063026 in *SPATA33* to be independently associated with AK risk on conditional analyses. Variants in both *DEF8* and *SPATA33* have been associated with pigmentation and cSCC[24,47]. Additionally, a previous phenome-wide association study identified a variant (rs258322) encoding cyclin-dependent kinase 10 (*CDK10*) that was associated with AK[49]. Of note, imputed expression levels of CDK10 were negatively correlated with risk allele dosage of SNP rs4268748[50]. A previous study proposed that variants at the *MC1R* locus regulate the *SPATA33* gene (in sun-exposed skin tissue) and the *CDK10* gene (in all tissues)[16]. Consistent with previous studies, our findings may provide evidence that the correlation of various genes with complicated linkage disequilibrium structures in the 16q24 region may collectively play a role in keratinocyte neoplasia. Future functional studies may help elucidate the role of AK-associated SNPs in this genomic region.

In our meta-analysis, an additional novel AK-susceptibility locus, *TRPS1* at 8q23, was identified. SNP rs7832568 lies inside an intronic region of *TRPS1*, which encodes a transcription factor that binds to a dynein light chain protein and suppresses the transcriptional activity of GATA regulated genes essential for bone and hair follicles, and is involved in sun sensitivity[16,51]. Interestingly, a recent genome-wide meta-analysis of cSCC identified a susceptibility SNP in the *TRPS1* locus[51]. The *TRPS1* (lead SNP rs7832568) locus will need to be validated in an external sample to confirm its implication in AK susceptibility.

Our gene-based analysis also identified the annexin A9 (*ANXA9*) gene on chromosome 1q21.3 as an AK risk locus. *ANXA9*, also known as pemphaxin, is targeted by pemphigus vulgaris antibodies in keratinocytes and may contribute to immune response and the acantholytic process[52,53]. In addition, a previous melanoma GWAS and a recent transcriptome-wide association study of cSCC identified *ANXA9* as a susceptibility locus[50,54]. The role of *ANXA9* in cutaneous carcinogenesis has been implicated by multi-omic methods and warrants further investigation.

This study's strengths include the robust sample sizes of both the discovery and validation cohorts and the associated comprehensive electronic health records derived from large, independent healthcare delivery systems. There are several limitations to be considered when interpreting the results. Our findings are limited to non-Hispanic white individuals, in which AKs almost exclusively arise, and results may not be extrapolated to individuals of non-European ancestry. This study defined AKs based on clinician-rendered diagnosis using the International Classification of Disease (ICD) diagnosis codes captured in the electronic healthcare systems. As such, we cannot exclude the possibility of undiagnosed AK arising in the controls. However, given the high reliability of AK codes[55], it is likely that the case definition has high validity.

In summary, this study provides independent replication of three previously reported AK susceptibility loci and identified seven novel loci contributing to AK pathophysiology. Our findings help elucidate pathways involved in AK pathophysiology and keratinocyte carcinogenesis, especially for immune regulation pathways, that we may benefit from targeted therapy. Identified loci could also serve as a framework for future functional investigations and as a basis for developing risk prediction models to identify individuals at high risk for keratinocyte neoplasia for future behavioral modification or chemoprevention trials.

## Materials and methods

**Case and control definition**. Potentially eligible cases in both cohorts were defined as participants who had a clinician-rendered AK diagnosis (International

Classification of Disease (ICD) diagnosis code version 9 of 702.0 and version 10 of L57.0) in the electronic health record. The control group included all participants without a relevant AK-ICD code. Our analysis included only self-reported NHW participants to minimize confounding risk due to ancestry differences. The Institutional Review Boards at the Kaiser Foundation Research Institute and the MGB Human Research Committee approved all study procedures.

**GERA cohort: genotyping, quality control, and imputation.** We report a GWAS of AK in 16,352 cases and 46,758 controls from the GERA cohort, NHW samples. The GERA cohort consists of 110,266 adults who consented to the Research Program on Genes, Environment, and Health at Kaiser Permanente Northern California (KPNC). GERA participants were genotyped at over 665,000 genetic markers on Affymetrix Axiom arrays optimized for individuals of European[56]. Samples with a sample call rate <0.97 have been filtered out. Standard quality control (QC) procedures were applied[57], with an additional step in which SNPs with a call rate <0.90 were removed. Detailed reports of genotyping and SNP quality control have been previously described[58]. Data then were pre-phased with SHAPE-IT v2.5[59]. SNPs were imputed from 1000 Genomes Project reference panel (phase I release, http://100genomes.org) using IMPUTE2 v2.3.1[60,61]. Additional QC procedures on genotyped data were applied before conducting GWAS[62]. The Genome Reference Consortium Human genome build 37 (GRCh37) was used in annotating variants. SNPs in the genotyped dataset were included in the imputed dataset that passed QC. We used the information $R^2$ from IMPUTE2 as a QC parameter, which estimates the imputed genotype's correlation to the actual genotype. Genetic markers with an imputation $R^2 > 0.7$ and minor allele frequency (MAF) > 0.01 were included in this study.

**GWAS analysis and covariate adjustment.** Logistic regression of AK for each SNP was performed using PLINK v1.9 (www.cog-genomics.org/plink/1.9/). We adjusted for age at cohort entry, sex, and top ten ancestry principal components (PCs). Principal component analysis (PCA) was performed using the smartpca program, part of the EIGENSOFT4.2 software package[63]. Details of the ancestry analyses are previously described[58]. We modeled data from each genetic marker using additive dosages accounting for the uncertainty of imputation. We defined the lead SNP as the most significant SNP within a 2 Mb (±1 Mb) window at each locus. Novel loci were defined as those located over 1 Mb apart from any previously reported locus.

**Conditional models.** We performed a stepwise procedure to explore independent signals within the loci identified in the GERA cohort[16]. Specifically, we fitted a new regression model in a 2 Mb (±1 Mb) window at each locus, including the top genome-wide significant SNP (smallest $P < 5 \times 10^{-8}$) at each locus identified in the association analysis step as a covariate (conditional model). We considered the top genome-wide significant SNP (smallest $P < 5 \times 10^{-8}$) at each locus identified from the conditional model as an independent signal and added it to the covariate list for the next iteration. A joint association of all the selected SNPs is iterated until no new genome-wide significant SNP at each locus remained associated. Conditional models were conducted using PLINK v1.9.

**Sensitivity analysis.** Given that AK and cSCC are reported to be genetically related[38], we performed a sensitivity analysis to explore the identified AK-associated signals among those without cSCC in the GERA discovery cohort. Details on cSCC case verification in GERA are described previously[24]. We excluded 7,121 subjects with at least one validated cSCC case (invasive or in situ), remaining 55,989 subjects in the sensitivity analysis (11,029 AK cases and 44,960 controls). Logistic regression of AK risk for each SNP was performed using PLINK v1.9 adjusting for age, sex, and top ten PCs.

**MGB Biobank cohort: genotyping, quality control, imputation, and GWAS analysis.** To validate the significant GERA-identified SNPs, we evaluated associations in the NHW subjects from the MGB Biobank, consisting of 5,110 AK cases and 24,020 controls. The MGB Biobank is an extensive integrated database containing clinical data from MGB HealthCare for ~100,000 consented patients and genomic data for over 35,000 participants[64]. MGB samples were genotyped using three versions of SNP array offered by Illumina (Illumina, Inc., San Diego, CA), including (1) Multi-Ethnic Genotyping Array (MEGA) array including 1,416,020 SNPs, (2) Expanded Multi-Ethnic Genotyping Array (MEGA Ex) array including 1,741,376 SNPs, and (3) Multi-Ethnic Global (MEG) array including 1,778,953 SNPs. GRCh37 has been used in the annotation of the variants. Imputation was performed using the Michigan Imputation Server that uses Minimac3[65]. MGB Biobank uses the HRC (Version r1.1 2016) reference panel for imputation. This HRC panel consists of 64,940 haplotypes of predominantly NHW ancestry. Haplotype phasing was performed using SHAPE-IT[59].

We included only NHW subjects, which were self-reported by patients, to minimize the risk for confounding due to ancestry differences and to be consistent with the discovery cohort. PCA was applied to characterize the population structure and exclude racial outliers. For the PCA, QC steps of genotyped data were conducted. Briefly, any variants with an SNP call rate <0.98 or MAF < 0.01, as well as any subjects with call rate <0.98, a discrepancy between the reported and predicted sex, evidence of an excess of homozygosity, or related or duplicated subjects (identity-by-descent [IBD] > 0.2) were excluded from the PCA.

For the genome-wide association analyses, imputed SNPs were used. Only common variants of three arrays (MEG, MEGA, MEGA EX) were included in all analyses after QCs. Specifically, an info score >0.8 (high-quality imputed SNPs), SNP call rates >0.95, and MAF > 0.01 were retained in the association analyses. PLINK 1.90 was used to conduct the genome-wide association analysis, adjusted for age, sex, and the top ten PCs. All phenotyping analyses were conducted using R (version 3.6.2, http://www.R-project.org/) and STATA 15.0 (StataCorp. 2017. Stata Statistical Software: Release 15. College Station, TX: StataCorp LLC).

**Replication of previously reported SNPs in GERA and MGB Biobank.** To assess whether the previously described AK-associated loci replicated in the GERA cohort, we tested three susceptibility SNPs identified in the previous GWAS with a genome-wide level of significance or after multiple testing corrections[28]. We reported the replication analysis results in the discovery and validation cohorts.

**Meta-analysis.** We conducted a meta-analysis of AK combining the discovery and validation cohorts using the PLINK software package (-meta-analysis). The analysis of each contributing GWAS had been performed independently, and quality control included assessments for population stratification in each data set. Variants that are commonly observed in both discovery and validation cohorts were included in the meta-analysis. A combined (discovery-validation) fixed-effect meta-analysis was performed using a Mantel–Haenzel method with the genome-wide P-value significance threshold set at $5 \times 10^{-8}$.

**Genes and biological pathway prioritization.** We calculated gene- and pathway-based p-values using the VEGAS2 software to prioritize genes and biological pathways[14,15]. A gene-based association analysis on the GWAS AK results of the GERA cohort was conducted using all variants assigned to a gene to compute gene-based P-value. Gene-based results were carried forward to run a pathway-based test. We analyzed the enrichment of the genes in 9,736 pathways or gene-sets (with 23,051 unique genes) derived from the Biosystem's database (https://vegas2.qimrberghofer.edu.au/biosystems20160324.vegas2pathSYM). All significant results using the Benjamini–Hochberg FDR control procedure with FDR 0.1 were presented[66].

**SNP-based heritability.** We used the LDSC software implemented in the LD Hub Web interface (http://ldsc.broadinstitute.org/ldhub/) for estimating array-heritability[67]. GWAS summary statistics of the GERA cohort were used to calculate heritability.

**Reporting summary.** Further information on research design is available in the Nature Research Reporting Summary linked to this article.

## Data availability

The GERA genotype data are available upon application to the KP Research Bank (https://researchbank.kaiserpermanente.org/). A subset of the GERA cohort consented for public use can be found in NIH/dbGaP: phs000674.v3.p3. The combined (GERA + MGB) meta-analysis GWAS summary statistics are available from the NHGRI-EBI GWAS Catalog (https://www.ebi.ac.uk/gwas/downloads/summary-statistics), study accession number GCST90095184.

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

## Acknowledgements

We thank Dr. Sae Kyu Lee for programming support, and Dr. Wenyu Song, Dr. Lu Chen Weng, and Dr. Hao Limin for statistical support. This work was supported by the National Institute of Arthritis and Musculoskeletal and Skin Diseases (K24 AR069760 to MA). Data used in this study were provided by the Kaiser Permanente Research Bank (KPRB) from the KPRB collection, which includes the Kaiser Permanente Research Program on Genes, Environment, and Health (RPGEH) and the Genetic Epidemiology Research on Adult Health and Aging (GERA), funded by the National Institutes of Health (RC2 AG036607), the Robert Wood Johnson Foundation, the Wayne and Gladys Valley Foundation, The Ellison Medical Foundation, and the Kaiser Permanente Community Benefits Program. H.C. and M.M.A. are also supported by the National Cancer Institute (NCI) R01CA2416323.

## Author contributions

Y.K. contributed to the conception of the study, acquisition, analysis, interpretation of data, and drafting and revising the work. J.Y. contributed to the analysis and interpretation of data and revising the work. H.H. contributed to the conception of the study, interpretation of data, and revising the work. E.J. contributed to the conception of the study, acquisition, interpretation of data, and revising the work. H.C. contributed to the conception of the study, interpretation of data, drafting, and revising the work. M.M.A. contributed to the conception of the study, acquisition, interpretation of data, drafting and revising the work, and supervision of the study.

## Competing interests

H.C. is an Editorial Board Member for Communications Biology, but was not involved in the editorial review of, nor the decision to publish this article. The authors declare no other competing interests.
