## [Peer Review File · Communications Biology]

Reviewers' comments:

Reviewer #1 (Remarks to the Author):

Kim et al conducted a GWAS of actinic keratosis (AK) in a discovery cohort of 63110 non-Hispanic white participants (16352 AK cases) with replication in a validation cohort of 29130 participants (5110 AK cases). The study confirmed several previously reported loci and identified eight novel loci, four of which replicated. A meta-analysis of the discovery and validation cohorts identified an additional locus. Genes in the areas in which the most significant SNPs were located were implicated in several pathways associated with AK susceptibility including pigmentation, immune regulation, and cell signaling and tissue remodeling.

Results

Lines 80-83 and Table 1: Given the substantial differences in age and sex distributions between cases and controls plus the much larger numbers of controls available for evaluation, why weren't controls matched to cases on age and sex? Matching would potentially reduce differential exposure distributions between cases and controls.

Lines 84-91: Were there any additional SNPs that showed evidence for association? Inclusion of a supplemental table showing the SNPs in each region would be helpful. Similarly, additional figures like supplemental figure 1 for the top loci would be informative and show the distribution of SNPs as well as additional genes in the top regions of interest.

Table 2: Was there evidence for linkage disequilibrium in the three SNPs on 20q11? Add location of the reported SNPs relative to the genes presented. Were SNPs exonic, within gene boundaries, or intergenic?

Line 97: What is SNP rs3506300?

Lines 98-100: How was evidence for there being multiple independent signals assessed?

Lines 109-112: Did the authors look at other SNPs in the regions of interest for supportive evidence of replication of association?

Discussion

Lines 163-170: What evidence is there that the functional SNPs related to the associations are in the genes emphasized by the authors? Were any studies done to assess functionality of the SNPs/genes presented?

Lines 297-304: The authors should discuss the limitations for this study.

Materials and Methods

Lines 316-319: Were controls required to have had dermatologic evaluation for both the discovery and replication cohorts? If not, a subset of controls could have had undiagnosed AK. The authors should discuss how this potential bias would influence the analyses.

Lines 327-330: What SNP arrays were used for the study? Were there any differences in arrays between participants who were classified as cases versus controls?

Line 378: What is 'familiar relationship'?

Lines 378-380: Two different call rate cutoffs are included. There appears to be repetition of the criteria (albeit different) used to exclude SNPs from association analysis.

Tables: several tables list 'Partners Biobank'. Others use MGB Biobank. The cohort name should be consistent throughout the manuscript.

Reviewer #2 (Remarks to the Author):

In this article, Yuhree et al present the results of a GWAS on actinic keratosis (AK) using health records from 63,110 non-Hispanic white participants of the Kaiser Permanente Genetic Epidemiology Research on Adult Health and Aging (GERA) cohort (discovery cohort), with replication in the Mass-General Brigham (MGB) Biobank (n= 29,130, validation cohort). The authors identified twelve loci ($p\text{-value} \leq 5.0E-8$), of which four replicated. In a meta-analysis (GERA+MGB), one additional locus was identified. Gene based analysis identified another locus, that was not identified in the GWAS discovery.

Major comments

The GWAS study is solid and validates previous findings on smaller cohorts, while identifying new variants. The health records are an advantage for large scale GWAS since allows for the collection of large series of cases and controls. Main comments are:

- 1) Individuals with AK are also likely to have field cancerization and multiple skin cancers. Given the overlapping genetic background between AK and skin cancer, how sure are the authors that the signals are due to AK and not to be driven squamous cell carcinoma (SCC) or basal cell carcinoma (BCC)? This is highlighted by the fact the FOXP1, HLA-DQA1 and RALY have been associated with SCC previously. In addition, in a recent publication (Eric Jorgenson et al Commun Biol . 2020) it was shown that 65% of people with cSCC had AK. The authors should perform a sensitivity analysis or stratified analysis on SCC status and look at the signals. Also, although different SNPs from the same gene (e.g.: FOXP1) were associated with either AK or SCC, it would be nice to see what was the p-value of the previously SCC- associated SNPs with AK and the pairwise LD between them.
- 2) In the abstract the authors mentioned 4 were replicated in the MGB cohort. The authors need to correct for multiple testing in the replication cohort ($0.05/12$) =0.004. This means that rs9271377 was not replicated (Table 2) in the MGB cohort.
- 3) I do not understand what were the new loci identified. In Table 2 of the manuscript the authors claimed that 12 loci were identified. This included the SNP rs55804368 mapped to the SLA2 gene, which was not replicated. In the meta-analysis the authors present the SNP rs73109224 from SOGA1. Could the authors explain what happened with the gene SLA2 in the meta-analysis? Furthermore, I do not agree with the authors that the signals from SOGA1 were replicated. The association of rs73109224 was not replicated in the partners biobank cohort ($p\text{-value} = 0.57$) and in the metaanalysis was 9.0×10^{-12} but was driven by the GERA cohort, which was not present in the discovery. Was this the SLA2 gene? Therefore I would not include this SNP.
- 4) The QQplots presented in the supplementary material suggest population stratification. Given the genetic diversity of the cohorts I suggest to include more PCs as covariates than the standard first four, unless that the authors can show that these 4 PCs account for most of the variation of the genetic ancestry. A QQ plot with 4 PCs and 10 PCs should help to clarify the extent of population stratification. Also, the authors did not mention if they only included north-European ancestry individuals in the replication cohort.
- 5) In addition from Table 4, it is clear that there is genetic heterogeneity in six of the associated SNPs in the meta-analysis (significant p-values from the Cochran Q statistics and I statistics). This is already an indication to use meta-analysis with random effects. Can the authors repeat the meta-analysis using a program that accounts for random effects?
- 6) The analysis of the genetic correlations is also misleading. Genetic correlations of 0.15 are not strong. I should not mention this as being correlated. What was the purpose of doing this analysis? The genetic correlations between AK and other skin cancers and pigmentation traits are more

relevant and underscore the shared genetic background between skin cancer and pigmentation. In this regard is rather puzzling that there was no correlation between AK and SCC, even though at gene level the overlap is high (9 genes out of 12) are shared between the two phenotypes). Could the authors explain this?

7) On page 6 the authors wrote that two signals 'are ~15 Kb apart and in linkage disequilibrium ($R^2 = 0.1773$, $D' = 0.8627$), suggesting that they can be correlated.' This sentence is not correct. Either they are in linkage equilibrium (meaning no correlation) or correlated (in LD). D' metric is not used anymore for assessing linkage disequilibrium in association analysis because it does not take into account sample size. Note the discrepancy between r^2 and D' . Which metric are the authors using to talk about LD? I would not use D' as a measure of strength of LD. In this respect an r^2 of 0.1 is not strong. For regions subjected to adaptive selection low LD between SNPs suggests more background LD due to adaptive selection than actual correlation due to disease status.

8) Conditional analysis is not clear from the methods. Which software did the authors use and what were the results? It would be nice for the reader to see this as supplementary information at least. The zoomplot does not really show independent signals as the authors claim in the manuscript

Minor comments

1) In the abstract the authors mentioned that they identified 8 new loci but in table 2 (and in the results) only 7 are underscored. Could the authors correct that? (Does this have to do with the disappearance of SLA2 gene from the table 4?)

2) SNP-based heritability estimate of 7.7% ($h^2 = 0.149$ SNP; SE=1.4%). Standard error should not be presented in percentage. Can the authors present the confidence interval?

3) Also, I am not sure how suitable is the database from LD hub from a mixed populations from USA. Not sure this is a representative dataset to use LD analysis. Don't the authors have individual genetic data that are more representative of the GERA cohort from other USA cohorts?

4) Which build is the annotation of the variants to gene based on? (which genome build)

5) Direction of effects of the cohorts in the meta-analysis should be added.

6) The authors should mention the limitations of the analysis including the use of electronic records in the discussion

7) Please use r^2 when referring to pair-wise LD. R^2 is reserved for imputation scores from mach software

Reviewer #3 (Remarks to the Author):

The study by Yuhyye et al. provides new insights on actinic keratosis- have identified 8 novel genetic variants, and 4 of them are replicated in an independent cohort. Most of them are pigmentation-related variants, however, this study improves the current understanding of the genetic basis of AK. study design, methods, and discussion points are reasonable. I have only few comments for the authors.

Materials and methods-

Is the control group free of AK only or any skin cancer 9 melanoma, keratinocyte cancer? Please specify

You have not specified the method of conditional analysis, included references

In discussion "Given melanin's role in protecting

284 keratinocytes from ultraviolet radiation-induced damage, which is the main environmental risk

285 factor for keratinocyte neoplasia formation, our identified hits in the pigmentation-related genes

286 appear to have clinical validity" - Does not make sense, either remove or explain more with references.

Several anthropometric traits, such as body mass index or fat mass and pack-years of
288 smoking, were inversely correlated with AK- IS pack-years of smoking an anthropometric trait?

Add footnotes for tables- explain abbreviations

Maryam Asgari, MD MPH
Professor of Dermatology
Massachusetts General Hospital
50 Staniford Street, Suite 230A
Boston, MA 02114

November 3, 2021

Re: Manuscript # COMMSBIO-21-1428-T "Genome-wide association study of actinic keratosis identifies novel risk loci implicated in pigmentation and immune regulation pathways."

Reviewers' comments:

Reviewer #1 (Remarks to the Author):

Kim et al conducted a GWAS of actinic keratosis (AK) in a discovery cohort of 63110 non-Hispanic white participants (16352 AK cases) with replication in a validation cohort of 29130 participants (5110 AK cases). The study confirmed several previously reported loci and identified eight novel loci, four of which replicated. A meta-analysis of the discovery and validation cohorts identified an additional locus. Genes in the areas in which the most significant SNPs were located were implicated in several pathways associated with AK susceptibility including pigmentation, immune regulation, and cell signaling and tissue remodeling.

Results

Lines 80-83 and Table 1: Given the substantial differences in age and sex distributions between cases and controls plus the much larger numbers of controls available for evaluation, why weren't controls matched to cases on age and sex? Matching would potentially reduce differential exposure distributions between cases and controls.

We thank the reviewer for his/her insightful comments and remarks. The reviewer is correct in noting differences in age and sex distributions between cases and controls. We agree with the reviewer that accounting for the differential distribution between cases and controls would be important. A recent review described the difficulty of matching of cases and controls given that population stratification is especially a challenge in large GWAS (Tam et al., *Nature Reviews Genetics*, 2019). Our logistic regression models were adjusted for age and sex, given that the covariate variables are independent of the SNPs, and age and sex are known to influence the AK risk. Also, we expected to reduce spurious associations and residual variance due to sampling artifacts or biases without losing the statistical power while controlling for these strong covariates.

Tam, Vivian, et al. "Benefits and limitations of genome-wide association studies." *Nature Reviews Genetics* 20.8 (2019): 467-484.

Lines 84-91: Were there any additional SNPs that showed evidence for association? Inclusion of a supplemental table showing the SNPs in each region would be helpful. Similarly, additional figures like supplemental figure 1 for the top loci would be informative and show the distribution of SNPs as well as additional genes in the top regions of interest.

We defined the lead SNP as the genome-wide significant top-associated SNP (smallest $P < 5 \times 10^{-8}$) within a 2 Mb (± 1 Mb) window at each locus. Per the reviewer's comment, we now have presented the regional plots for each identified locus (Supplementary Figures 2a-2f). This has been added in the 6th paragraph of the Results section and now reads as follows:

"...We generated a regional association plot at the novel AK-susceptibility loci to illustrate the signals at this genomic region (Supplementary Figure 2a-2f)..."

Table 2: Was there evidence for linkage disequilibrium in the three SNPs on 20q11? Add location of the reported SNPs relative to the genes presented. Were SNPs exonic, within gene boundaries, or intergenic?

There was evidence for linkage disequilibrium in the three SNPs on 20q11 (rs6059655 in *RALY*, rs2425025 in *MMP24*, and rs73109224 in *SOGAI*). The strongest correlation with r^2 statistics of 0.8522 was noted between rs6059655 and rs2425025, and 0.2487 for rs6059655 and rs73109224 and 0.3074 for rs2425025 and rs73109224. SNPs rs6059655, rs2425025, and rs73109224 are intron variants of genes *RALY*, *MMP24*, and *SOGAI*, respectively. Given that we have removed SNPs rs55804379 and rs73109224 from the reported SNP list per the reviewer's comment (Reviewer #2 Comment #3), only SNPs rs6059655 in *RALY* and rs2425025 in *MMP24* are now reported in the revised manuscript. Per the reviewer's comment, this has been described in the 5th paragraph of the Results section. The section now reads as follows:

"...SNPs rs6059655 in *RALY* and rs2425025 in *MMP24* on 20q11 are in linkage disequilibrium ($r^2 = 0.85$)..."

Line 97: What is SNP rs3506300?

We thank the reviewer for noting the typo. We have fixed the error in the 3rd paragraph of the Results section, and it now reads as follows:

"...These two SNPs (rs35063026 and rs1805008) are ~0.25 Mb apart and in linkage equilibrium ($r^2 = 0.005$), suggesting that they are independent signals in the same locus..."

Lines 98-100: How was evidence for there being multiple independent signals assessed?

To identify independent signals within each locus, we used a stepwise procedure using conditional models. This method has been addressed in the 5th paragraph of the Methods, and we added a reference (Visconti et al., Nature Communications 2018). The section reads as follows:

“...We performed a stepwise procedure to explore independent signals within the loci identified in the GERA cohort.¹³ Specifically, we fitted a new regression model in a 2 Mb (± 1 Mb) window at each locus, including the genome-wide significant top-associated SNP (smallest $P < 5 \times 10^{-8}$) identified in the association analysis step as a covariate (conditional model). We considered the genome-wide significant top-associated SNP identified from the conditional model as an independent signal and added it to the covariate list for the next iteration. A joint association of all the selected SNPs is iterated until no new genome-wide significant SNP at each locus remained associated. Conditional models were conducted using PLINK v1.9...”

Lines 109-112: Did the authors look at other SNPs in the regions of interest for supportive evidence of replication of association?

For the purpose of replication of association, we only looked at the genome-wide significant lead SNPs of each locus. Per the reviewer's comment, we looked at other SNPs in the susceptibility loci. There were no SNPs in the susceptibility loci that reached a genome-wide significance level in the replication cohorts.

Discussion

Lines 163-170: What evidence is there that the functional SNPs related to the associations are in the genes emphasized by the authors? Were any studies done to assess functionality of the SNPs/genes presented?

We appreciate the reviewer's important comment. The AK-associated SNPs were mapped to genes by physical position on the genome (positional mapping). There have been no studies on functional mapping and annotation of genetic associations for AK to date to the best of our knowledge. However, several functionality studies of the SNPs/genes associated with pigmentation have been reported, which is a risk factor for AK and skin cancer. We have identified multiple SNPs within genes involved in human pigmentation traits. These include *SLC45A2* in 5p13, *IRF4* in 6p25, *OCA2/HERC2* in 15q13, and *MC1R* in 16q24. For example, SNP rs16891982 in the *SLC45A2* gene, which encodes a transporter protein that mediates melanin synthesis, correlates with reduced melanin content in cultured human melanocytes (Spichenok et al. *Forensic Science International Genetics* 2011). SNP rs12203592 in the *IRF4* gene modulates enhancer-mediated transcriptional regulation and physically interacts with the *IRF4* gene promoter. *IRF4* cooperates with the melanocyte master regulator MITF to activate the tyrosinase expression that catalyzes melanin production (Visser et al. *Human Molecular Genetics* 2014). SNP rs12350739 and the highly conserved surrounding region function as enhancers regulating *BNC2* transcription in human melanocytes (Visser et al. *Human Molecular Genetics* 2014). The SNP rs1126809 variant of tyrosinase may cause changes at the post-translational modification site, leading to dysregulation of melanin synthesis within the melanosomes (Khoruddin et al. *Scientific reports* 2021). In addition, pigmentation genes on 16q24 are thought to be complicated by their proximity to adjacent

genes in the pigmentation pathways. Per the reviewer's comment, these have been further described in the 2nd paragraph of the Discussion section and the section now reads as follows:

“...and it may cause changes at the post-translational modification site, leading to dysregulation of melanin synthesis within the melanosomes...”

“...Similarly, SNP rs16891982 lies in *SLC45A2*, which encodes a transporter protein that mediates melanin synthesis, correlates with reduced melanin content in cultured human melanocytes...”

“...In addition, SNP rs12350739, an intergenic SNP of basophilin 2 (*BNC2*), and the highly conserved surrounding region function as enhancers regulating *BNC2* transcription in human melanocytes...”

Spichenok, Olga, et al. "Prediction of eye and skin color in diverse populations using seven SNPs." *Forensic Science International: Genetics* 5.5 (2011): 472-478.

Visser, Mijke, Robert-Jan Palstra, and Manfred Kayser. "Allele-specific transcriptional regulation of IRF4 in melanocytes is mediated by chromatin looping of the intronic rs12203592 enhancer to the IRF4 promoter." *Human molecular genetics* 24.9 (2015): 2649-2661.

Visser, Mijke, Robert-Jan Palstra, and Manfred Kayser. "Human skin color is influenced by an intergenic DNA polymorphism regulating transcription of the nearby *BNC2* pigmentation gene." *Human molecular genetics* 23.21 (2014): 5750-5762.

Khoruddin, Nurul Ain, et al. "Pathogenic nsSNPs that increase the risks of cancers among the Orang Asli and Malays." *Scientific reports* 11.1 (2021): 1-22.

Lines 297-304: The authors should discuss the limitations for this study.

Per the reviewer's suggestion, we have clarified the limitations of this study, and the 8th paragraph of the discussion has been revised. The section now reads as follows:

“...There are several limitations to be considered when interpreting the results. Our findings are limited to non-Hispanic white individuals, in which AKs almost exclusively arise, and results may not be extrapolated to individuals of non-European ancestry. This study defined AKs based on clinician-rendered diagnosis using the International Classification of Disease (ICD) diagnosis codes captured in the electronic healthcare systems, and as such, we cannot exclude the possibility of undiagnosed AK arising in the controls. However, given the high reliability of AK codes,⁵² it is likely that the case definition has high validity...”

Materials and Methods

Lines 316-319: Were controls required to have had dermatologic evaluation for both the discovery

and replication cohorts? If not, a subset of controls could have had undiagnosed AK. The authors should discuss how this potential bias would influence the analyses.

We thank the reviewer for this important comment. The AK cases were defined as physician-rendered diagnoses and captured using the ICD-9 or ICD-10 diagnosis codes in the electronic health systems; dermatologic evaluation was not required to define the cases or controls. The reviewer is correct in noting that the controls, especially in the MGB cohort, may include undetected or undiagnosed AK. However, undiagnosed AK cases in the GERA cohort may be limited where most individuals are followed in the health care system; hence, we do not expect it to alter the significance of our findings. This issue has been discussed in the 8th paragraph of the Discussion, and the section now reads as follows:

“...This study defined AKs based on clinician-rendered diagnosis using the International Classification of Disease (ICD) diagnosis codes captured in the electronic healthcare systems, and as such, we cannot exclude the possibility of undiagnosed AK arising in the controls...”

Lines 327-330: What SNP arrays were used for the study? Were there any differences in arrays between participants who were classified as cases versus controls?

To genotype the non-Hispanic white GERA individuals, the Affymetrix Axiom array which was designed for European ancestry individuals was used (Hoffman et al. *Genomics* 2011). MGB samples were genotyped using three versions of SNP array offered by Illumina, Multi-Ethnic Genotyping Array (MEGA), Expanded Multi-Ethnic Genotyping Array (MEGA Ex), and Multi-Ethnic Global (MEG) array. There are no differences in arrays between GERA non-Hispanic White cases and control given that same array used in GERA. In the MGB cohort, the proportion of each array in the cases (MEG 72.9%, MEGA 14.1%, and MEGA EX 13.0%) are comparable to that in the controls (MEG 68.1%, MEGA 17.3%, and MEGA EX 14.6%).

Hoffmann, Thomas J., et al. "Next generation genome-wide association tool: design and coverage of a high-throughput European-optimized SNP array." *Genomics* 98.2 (2011): 79-89.

Line 378: What is "familiar relationship"?

We excluded subjects with evidence of relatedness (i.e., 2nd degree relatives) to ensure that the subjects included in the association analysis were all unrelated. We have removed one of a pair if identity-by-descent [IBD] > 0.2. We have clarified this in the 3rd paragraph of the Methods section, and it now reads as follows:

“...Briefly, any variants with an SNP call rate < 0.98 or MAF < 0.01, as well as any subjects with call rate < 0.98, a discrepancy between the reported and predicted sex, evidence of an excess of homozygosity, or related or duplicated subjects (identity-by-descent [IBD] > 0.2) were excluded from the PCA...”

Lines 378-380: Two different call rate cutoffs are included. There appears to be repetition of the criteria (albeit different) used to exclude SNPs from association analysis.

We thank the reviewer for pointing this out. The call rate cutoffs were clarified in the 8th and 9th paragraphs in the 8th and 9th paragraphs of the Methods, and now the section reads as follows:

“...Briefly, any variants with an SNP call rate < 0.98 or MAF < 0.01 , as well as any subjects with call rate < 0.98 , a discrepancy between the reported and predicted sex, evidence of an excess of homozygosity, or related or duplicated subjects (identity-by-descent [IBD] > 0.2) were excluded from the PCA...”

“...For the genome-wide association analyses, imputed SNPs were used. Only common variants of three arrays (MEG, MEGA, MEGA EX) were included in all analyses after QCs. Specifically, an info score > 0.8 (high-quality imputed SNPs), SNP call rates > 0.95 , and MAF > 0.01 were retained in the association analyses. ...”

Tables: several tables list "Partners Biobank". Others use MGB Biobank. The cohort name should be consistent throughout the manuscript.

Thank you for pointing this out. We updated the labels in the tables to be consistent throughout the manuscript.

Reviewer #2 (Remarks to the Author):

In this article, Yuhree et al present the results of a GWAS on actinic keratosis (AK) using health records from 63,110 non-Hispanic white participants of the Kaiser Permanente Genetic Epidemiology Research on Adult Health and Aging (GERA) cohort (discovery cohort), with replication in the Mass-General Brigham (MGB) Biobank (n= 29,130, validation cohort). The authors identified twelve loci (p-value $\leq 5.0E-8$), of which four replicated. In a meta-analysis (GERA+MGB), one additional locus was identified. Gene based analysis identified another locus, that was not identified in the GWAS discovery.

Major comments

The GWAS study is solid and validates previous findings on smaller cohorts, while identifying new variants. The health records are an advantage for large scale GWAS since allows for the collection of large series of cases and controls. Main comments are:

1) Individuals with AK are also likely to have field cancerization and multiple skin cancers.

Given the overlapping genetic background between AK and skin cancer, how sure are the authors that the signals are due to AK and not to be driven squamous cell carcinoma (SCC) or basal cell carcinoma (BCC)? This is highlighted by the fact the FOXP1, HLA-DQA1 and RALY have been associated with SCC previously. In addition, in a recent publication (Eric Jorgenson et al Commun

Biol . 2020) it was shown that 65% of people with cSCC had AK. The authors should perform a sensitivity analysis or stratified analysis on SCC status and look at the signals.

We thank the reviewer for his/her insightful comments and remarks. The reviewer is correct in noting that AK is highly associated with skin cancer, especially SCC. They may share genetic factors evidenced by multiple common susceptibility loci and environmental factors such as ultraviolet (UV) exposure. For those polygenic diseases caused by the combined action of more than one gene that shares common risk factors, we expect that there may be natural genetic overlaps. For example, *IRF4* has been associated with not only AK and skin cancers, including SCC, BCC, and melanoma, but also with pigmentation, hair and eye color, and sun sensitivity. We agree with the reviewer that further elaborating on this point using sensitivity analysis would be helpful. However, such an analysis is beyond the scope of our paper, which aims to identify associations of SNPs with AK by testing for differences in the allele frequency of genetic variants between ancestrally similar individuals (non-Hispanic Whites) who differ phenotypically (AK cases vs. controls). For this reason, we sought to identify the SNPs associated with AK risk and did not exclude subjects with skin cancer history. Nevertheless, we carefully considered the reviewer's astute comment, and we have added the following sentence to the 2nd paragraph in the Discussion:

“...Further studies including pathway analysis of identified variants and genes as well as functional studies may help elucidate the association between AK and skin cancers...”

Also, although different SNPs from the same gene (e.g.: FOXPI) were associated with either AK or SCC, it would be nice to see what was the p-value of the previously SCC- associated SNPs with AK and the pairwise LD between them.

The SNP rs62246017 was previously associated with SCC (Asgari et al. *JID* 2016), and the p-value was 1.16×10^{-8} . SNPs rs62246017 and rs7638354 were correlated with r^2 of 0.882. Per the reviewer's comment, this has been described in the 3rd paragraph of the Discussion, and the section now reads as follows:

“...The SNP rs62246017 in *FOXPI* was previously associated with cSCC, and it was in linkage disequilibrium with rs7638354 identified in the current study ($r^2=0.882$)...”

Asgari, Maryam M., et al. "Identification of susceptibility loci for cutaneous squamous cell carcinoma." *Journal of Investigative Dermatology* 136.5 (2016): 930-937.

2) In the abstract the authors mentioned 4 were replicated in the MGB cohort. The authors need to correct for multiple testing in the replication cohort (0.05/12) =0.004. This means that rs9271377 was not replicated (Table 2) in the MGB cohort.

The observation is correct. We have reiterated the replication results in the 4th paragraph of the Results section and now reads as follows:

“...Three additional SNPs replicated at Bonferroni significance ($P < 0.004 = 0.05/12$; 12 SNPs tested) including rs12350739 in *BNC2*, rs6059655 in *RALY*, and rs2425025 in *MMP24*. The remaining loci (rs62247035 in *FOXP1*, rs9271377 in *HLA-DQA1*, rs12916300 in *HERC2*) did not reach statistical significance, although their direction of effect was consistent with those of the discovery cohort...”

3) I do not understand what were the new loci identified. In Table 2 of the manuscript the authors claimed that 12 loci we identified. This included the SNP rs55804368 mapped to the SLA2 gene, which was not replicated. In the meta-analysis the authors present the SNP rs73109224 from SOGA1. Could the authors explain what happened with the gene SLA2 in the meta-analysis? Furthermore, I do not agree with the authors that the signals from SOGA1 were replicated. The association of rs73109224 was not replicated in the partners biobank cohort (p-value =0.57) and in the metaanalysis was 9.0 e-12 but was driven by the GERA cohort, which was not present in the discovery. Was this the Sla2 gene? Therefore I would not include this SNP.

We appreciate the reviewer’s important comment. The reviewer is correct in noting that SNP rs55804368 in the *SLA2* gene identified in the GERA cohort was not replicated in the MGB cohort and not identified in the meta-analysis. We defined the lead SNP as the most significant SNP within a 2 Mb (± 1 Mb) window at each locus. SNP rs55804368 in the *SLA2* gene reached genome-wide significance in the meta-analysis (OR 1.16, $P = 1.81 \times 10^{-9}$), but it was not presented as a lead SNP given that SNP rs73109224 in the *SOGA1* gene had smaller P-value and they were in the 2-Mb window. As per the reviewer's comment, we have removed SNPs rs55804379 and rs73109224 from the reported SNP list. These results are updated in the Abstract and the 2nd paragraph of the Results sections. We have modified Table 2, Table 4, and Figure 3. The corresponding sections are now read as follow:

“...Genes within the identified loci are implicated in pigmentation (*SLC45A2*, *IRF4*, *BNC2*, *TYR*, *DEF8*, *RALY*, *HERC2*, and *TRPS1*), immune regulation (*FOXP1* and *HLA-DQA1*), and cell signaling and tissue remodeling (*MMP24*) pathways...”

“...These included *FOXP1* (lead SNP rs62247035), *SLC45A2* (lead SNP rs16891982), *HLA-DQA1* (lead SNP rs9271377), *BNC2* (lead SNP rs12350739), *RALY* (lead SNP rs6059655), and *MMP24* (lead SNP rs2425025)....”

4) The QQplots presented in the supplementary material suggest population stratification. Given the genetic diversity of the cohorts I suggest to include more PCs as covariates that the standard first four, unless that the authors can show that this 4 PCs account for most of the variation of the genetic ancestry. A QQ plot with 4PCs and 10 PCs should help to clarify the extend of population stratification. Also, the authors did not mention if they only included north-European ancestry individuals in the replication cohort

We thank the reviewer for allowing us to clarify the methods. We have adjusted for the top 10 PCs in our association analyses in both discovery and replication cohorts. Also, only non-Hispanic White

individuals were included in both cohorts. As per the reviewer's comment, we now clarify this in the 4th, 8th, and 9th paragraphs of the Methods section. This section now reads as follows:

“...We adjusted for age at cohort entry, sex, and top ten ancestry principal components (PCs)...”

“...We included only NHW subjects, which were self-reported by patients, to minimize the risk for confounding due to ancestry differences. A principal components analysis (PCA) was applied to characterize population structure and exclude racial outliers...”

“...PLINK 1.90 was used to conduct the genome-wide association analysis, adjusted for age, sex, and the top ten PCs...”

5) In addition from Table 4, it is clear that there is genetic heterogeneity in six of the associated SNPs in the meta-analysis (significant p-values from the Cochran Q statistics and I statistics). This is already an indication to use meta-analysis with random effects. Can the authors repeat the meta-analysis using a program that accounts for random effects?

We thank the reviewer for pointing this out. We have accounted for random effects in the meta-analysis, per the reviewer's recommendation; we added the random-effects meta-analysis P values to Table 4.

6) The analysis of the genetic correlations is also misleading. Genetic correlations of 0.15 are not strong. I should not mention this as being correlated. What was the purpose of doing this analysis? the genetic correlations between AK and other skin cancers and pigmentation traits are more relevant and underscore the shared genetic background between skin cancer and pigmentation. In this regard is rather puzzling that there was no correlation between AK and SCC, even though at gene level the overlap is high (9 genes out of 12) are shared between the two phenotypes). Could the authors explain this?

We thank the reviewer for this important comment. We agree with the reviewer that the results from the genetic correlations may be obscure, given the lack of correlation between AK and SCC. Unfortunately, only self-reported SCC diagnosis, not validated SCC cases, was available to analyze the genetic correlation in the LDHub, and there might be underestimation or misclassification of the SCC diagnosis. As per the reviewer's comment, we have removed the genetic correlations from the manuscript and revised the manuscript to focus on association analysis and pathways analysis.

7) On page 6 the authors wrote that two signals "are ~15 Kb apart and in linkage disequilibrium ($R^2_{119} = 0.1773$, $D'D' = 0.8627$), suggesting that 120 they can be correlated." This sentence is not correct. Either they are in linkage equilibrium (meaning no correlation) or correlated (in LD). $D'D'$ metrics is not used anymore for assessing linkage disequilibrium in association analysis because it does not take into account sample size. Note the discrepancy between r^2 and D' . Which metric are

the authors using to talk about LD? I would not use D' as a measure of strength of LD. In this respect an r² of 0.1 is not strong. For regions subjected to adaptive selection low LD between SNPs suggests more background LD due to adaptive selection than actual correlation due to disease status.

We appreciate the reviewer's insightful comment. We now removed the D' metrics from the manuscript and revised the manuscript accordingly.

8) Conditional analysis is not clear from the methods. Which software did the authors use and what were the results? It would be nice for the reader to see this as supplementary information at least. The zoomplot does not really show independent signals as the authors claim in the manuscript

We thank the reviewer for allowing us to clarify the conditional analysis. This has been reiterated in the 5th paragraph of the Methods section, including software, and we have added a reference (Visconti et al. *Nature Communications* 2018). The section now reads as follows:

“... We performed a stepwise procedure to explore independent signals within the loci that were identified in the GERA cohort. Specifically, we fitted a new regression model in a 2 Mb (\pm 1 Mb) window at each locus, including the genome-wide significant top-associated SNP (smallest $P < 5 \times 10^{-8}$) identified in the association analysis step as a covariate (conditional model). We considered the genome-wide significant top-associated SNP identified from the conditional model as an independent signal and added it to the covariate list for the next iteration. A joint association of all the selected SNPs is iterated until no new genome-wide significant SNP at each locus remained associated. Conditional models were conducted using PLINK v1.9...”

Visconti, Alessia, et al. "Genome-wide association study in 176,678 Europeans reveals genetic loci for tanning response to sun exposure." *Nature communications* 9.1 (2018): 1-7.

Minor comments

1) In the abstract the authors mentioned that they identified 8 new loci but in table 2 (and in the results) only 7 are underscored. Could the authors correct that? (Does this have to do with the disappearance of SLA2 gene from the table 4?)

We thank the reviewer for pointing this out. The reviewer is correct in noting that seven novel loci, not eight, were identified in the discovery cohort in previous Table 2. Per the reviewer's suggestion (Reviewer #2 Major comment #3), we have removed SNPs rs55804379 and rs73109224 from the manuscript. We clarified the number of newly identified AK-susceptibility loci in the Abstract, the 2nd paragraph of the Results, and the Discussion sections, and these read as follow:

“...We identified eleven loci ($P < 5 \times 10^{-8}$), including six novel loci, of which three replicated. In a meta-analysis (GERA+MGB), one additional locus was identified...”

“...We identified eleven genome-wide significant ($P < 5 \times 10^{-8}$) loci associated with AK (Table 2 and Figure 1), of which six loci have not been previously reported...”

“...In summary, this study provided the first independent replication of three previously reported AK susceptibility loci and identified seven novel loci contributing to AK pathophysiology...”

2) SNP-based heritability estimate of 7.7% (h^2 149 SNP; SE=1.4%). Standard error should not be presented in percentage. Can the authors present the confidence interval?

Thank you for the comment. We have presented the confidence interval for the heritability estimates in the 8th paragraph of the Results section. The section now reads as follows:

“...We estimated SNP-based heritability in the KPNC NHW cohort using linkage disequilibrium score regression (LDSR) software on the LDHub website (<http://ldsc.broadinstitute.org/ldhub/>), and we found a SNP-based heritability estimate of 0.077 (h^2_{SNP} ; 95% CI 0.05-0.10)...”

3) Also, I am not sure how suitable is the database from LD hub from a mixed populations from USA. Not sure this is a representative dataset to use LD analysis. Don't the authors have individual genetic data that are more representative of the GERA cohort from other USA cohorts?

We appreciate the reviewer's insightful comment. We have limited the correlation analysis from LDhub to the European population, given that we only included non-Hispanic Whites in both GERA and MGB cohorts. To increase clarity, per the reviewers' comment, we have revised the manuscript focusing on the association analysis and pathways analysis and removed the genetic correlation analysis performed on LD Hub.

4) Which build is the annotation of the variants to gene based on? (which genome build)

The Genome Reference Consortium Human genome build 37 (GRCh37) has been used in the annotation of the variants. Per the reviewer's comment, we have explained this in the 2nd paragraph of the Methods section, and the new sentence reads as follows:

“...The Genome Reference Consortium Human genome build 37 (GRCh37) has been used in the annotation of the variants...”

5) Direction of effects of the cohorts in the meta-analysis should be added.

We have added the direction of the effects of the two cohorts in Table 4.

6) The authors should mention the limitations of the analysis including the use of electronic records in the discussion

We thank the reviewer for this comment. The limitations of the analysis, including the use of electronic health records, have been further discussed in the 8th paragraph of the Discussion, and the section reads as follows:

“...There are several limitations to be considered when interpreting the results. Our findings are limited to non-Hispanic white individuals, in which AKs almost exclusively arise, and results may not be extrapolated to individuals of non-European ancestry. This study defined AKs based on clinician-rendered diagnosis using the International Classification of Disease (ICD) diagnosis codes captured in the electronic healthcare systems, and as such, we cannot exclude the possibility of undiagnosed AK arising in the controls. However, given the high reliability of AK codes,⁵² it is likely that the case definition has high validity. ...”

7) Please use r^2 when referring to pair-wise LD. R^2 is reserved for imputation scores from mach software

We now have used r^2 when referring to pair-wise linkage disequilibrium, while R^2 corresponds to imputation scores.

Reviewer #3 (Remarks to the Author):

The study by Yuhyye et al. provides new insights on actinic keratosis- have identified 8 novel genetic variants, and 4 of them are replicated in an independent cohort. Most of them are pigmentation-related variants, however, this study improves the current understanding of the genetic basis of AK. study design, methods, and discussion points are reasonable. I have only few comments for the authors.

Materials and methods-

Is the control group free of AK only or any skin cancer 9 melanoma, keratinocyte cancer? Please specify

We thank the reviewer for his/her insightful comments and remarks. The control group is subjects who were genotyped and were not diagnosed AK. We did not exclude subjects with any skin cancer to avoid any selection bias.

You have not specified the method of conditional analysis, included references

Per the reviewer's comment, we have reiterated the methods that were used for conditional analysis in the Methods and added a reference (Visconti et al., Nature Communications 2018). The section reads as follows:

“...We performed a stepwise procedure to explore independent signals within the loci identified in the GERA cohort. Specifically, we fitted a new regression model in a 2 Mb (\pm 1 Mb) window at each locus, including the genome-wide significant top-associated SNP (smallest $P < 5 \times 10^{-8}$) identified in the association analysis step as a covariate (conditional model). We considered the genome-wide significant top-associated SNP (smallest $P < 5 \times 10^{-8}$) identified from the conditional model as an independent signal and added it to the covariate list for the next iteration. A joint association of all the selected SNPs is iterated until no new genome-wide significant SNP at each locus remained associated. Conditional models were conducted using PLINK v1.9...”

Visconti, Alessia, et al. "Genome-wide association study in 176,678 Europeans reveals genetic loci for tanning response to sun exposure." Nature communications 9.1 (2018): 1-7.

In discussion "Given melanin's role in protecting

***284 keratinocytes from ultraviolet radiation-induced damage, which is the main environmental risk
285 factor for keratinocyte neoplasia formation, our identified hits in the pigmentation-related genes
286 appear to have clinical validity" - Does not make sense, either remove or explain more with
references.***

We have removed the paragraph from the discussion.

Several anthropometric traits, such as body mass index or fat mass and pack-years of

288 smoking, were inversely correlated with AK- IS pack-years of smoking an anthropometric trait?

We thank the reviewer for pointing this out. To increase the clarity of our manuscript, per the reviewers' comments, we have removed the correlation analysis from the manuscript to focus on the analysis from the association analysis and pathways analysis to avoid any confusion.

Add footnotes for tables- explain abbreviations

We have added footnotes for tables to explain the abbreviations.

Thank you in advance for your further consideration of our manuscript.

Sincerely,

Maryam M. Asgari

Maryam M. Asgari, MD MPH
Department of Dermatology
masgari@partners.org

From the editor:

“Although we deeply appreciate the revisions made in response to the reviewers, unfortunately we do not believe that these will be sufficient to satisfy the reviewers on this occasion. Before sending the manuscript back to them, we therefore ask you to perform the sensitivity analysis as suggested by Referee #2, as follows: “

1) Individuals with AK are also likely to have field cancerization and multiple skin cancers. Given the overlapping genetic background between AK and skin cancer, how sure are the authors that the signals are due to AK and not to be driven squamous cell carcinoma (SCC) or basal cell carcinoma (BCC)? This is highlighted by the fact the FOXP1, HLA-DQA1 and RALY have been associated with SCC previously. In addition, in a recent publication (Eric Jorgenson et al Commun Biol . 2020) it was shown that 65% of people with cSCC had AK. The authors should perform a sensitivity analysis or stratified analysis on SCC status and look at the signals.

Maryam Asgari, MD MPH
Professor of Dermatology
Massachusetts General Hospital
50 Staniford Street, Suite 230A
Boston, MA 02114

December 3, 2021

Re: Manuscript # COMMSBIO-21-1428-T "Genome-wide association study of actinic keratosis identifies novel risk loci implicated in pigmentation and immune regulation pathways."

Reviewers' comments:

Reviewer #1 (Remarks to the Author):

Kim et al conducted a GWAS of actinic keratosis (AK) in a discovery cohort of 63110 non-Hispanic white participants (16352 AK cases) with replication in a validation cohort of 29130 participants (5110 AK cases). The study confirmed several previously reported loci and identified eight novel loci, four of which replicated. A meta-analysis of the discovery and validation cohorts identified an additional locus. Genes in the areas in which the most significant SNPs were located were implicated in several pathways associated with AK susceptibility including pigmentation, immune regulation, and cell signaling and tissue remodeling.

Results

Lines 80-83 and Table 1: Given the substantial differences in age and sex distributions between cases and controls plus the much larger numbers of controls available for evaluation, why weren't controls matched to cases on age and sex? Matching would potentially reduce differential exposure distributions between cases and controls.

We thank the reviewer for his/her insightful comments and remarks. The reviewer is correct in noting differences in age and sex distributions between cases and controls. We agree with the reviewer that accounting for the differential distribution between cases and controls would be important. A recent review described the difficulty of matching of cases and controls given that population stratification is especially a challenge in large GWAS (Tam et al., *Nature Reviews Genetics*, 2019). Our logistic regression models were adjusted for age and sex, given that the covariate variables are independent of the SNPs, and age and sex are known to influence the AK risk. Also, we expected to reduce spurious associations and residual variance due to sampling artifacts or biases without losing the statistical power while controlling for these strong covariates.

Tam, Vivian, et al. "Benefits and limitations of genome-wide association studies." *Nature Reviews Genetics* 20.8 (2019): 467-484.

Lines 84-91: Were there any additional SNPs that showed evidence for association? Inclusion of a supplemental table showing the SNPs in each region would be helpful. Similarly, additional figures like supplemental figure 1 for the top loci would be informative and show the distribution of SNPs as well as additional genes in the top regions of interest.

We defined the lead SNP as the genome-wide significant top-associated SNP (smallest $P < 5 \times 10^{-8}$) within a 2 Mb (± 1 Mb) window at each locus. Per the reviewer's comment, we now have presented the regional plots for each identified locus (Supplementary Figures 2a-2f). This has been added in the 7th paragraph of the Results section and now reads as follows:

"...Regional association plots at the novel AK-susceptibility loci are presented in Supplementary Figure 2a-2f..."

Table 2: Was there evidence for linkage disequilibrium in the three SNPs on 20q11? Add location of the reported SNPs relative to the genes presented. Were SNPs exonic, within gene boundaries, or intergenic?

There was evidence for linkage disequilibrium in the three SNPs on 20q11 (rs6059655 in *RALY*, rs2425025 in *MMP24*, and rs73109224 in *SOGAI*). The strongest correlation with r^2 statistics of 0.8522 was noted between rs6059655 and rs2425025, and 0.2487 for rs6059655 and rs73109224 and 0.3074 for rs2425025 and rs73109224. SNPs rs6059655, rs2425025, and rs73109224 are intron variants of genes *RALY*, *MMP24*, and *SOGAI*, respectively. Given that we have removed SNPs rs55804379 and rs73109224 from the reported SNP list per the reviewer's comment (Reviewer #2 Comment #3), only SNPs rs6059655 in *RALY* and rs2425025 in *MMP24* are now reported in the revised manuscript. Per the reviewer's comment, this has been described in the 2nd paragraph of the Results section. The section now reads as follows:

"...SNPs rs6059655 in *RALY* and rs2425025 in *MMP24* on 20q11 are in linkage disequilibrium ($r^2 = 0.85$)..."

Line 97: What is SNP rs3506300?

We thank the reviewer for noting the typo. We have fixed the error in the 3rd paragraph of the Results section, and it now reads as follows:

"...These two SNPs (rs35063026 and rs1805008) are ~0.25 Mb apart and in linkage equilibrium ($r^2 = 0.005$), suggesting that they are independent signals in the same locus..."

Lines 98-100: How was evidence for there being multiple independent signals assessed?

To identify independent signals within each locus, we used a stepwise procedure using conditional models. This method has been addressed in the 4th paragraph of the Methods, and we added a reference (Visconti et al., Nature Communications 2018). The section reads as follows:

“...We performed a stepwise procedure to explore independent signals within the loci identified in the GERA cohort.¹³ Specifically, we fitted a new regression model in a 2 Mb (\pm 1 Mb) window at each locus, including the genome-wide significant top-associated SNP (smallest $P < 5 \times 10^{-8}$) identified in the association analysis step as a covariate (conditional model). We considered the genome-wide significant top-associated SNP identified from the conditional model as an independent signal and added it to the covariate list for the next iteration. A joint association of all the selected SNPs is iterated until no new genome-wide significant SNP at each locus remained associated. Conditional models were conducted using PLINK v1.9...”

Lines 109-112: Did the authors look at other SNPs in the regions of interest for supportive evidence of replication of association?

For the purpose of replication of association, we only looked at the genome-wide significant lead SNPs of each locus. Per the reviewer's comment, we looked at other SNPs in the susceptibility loci. There were no SNPs in the susceptibility loci that reached a genome-wide significance level in the replication cohorts.

Discussion

Lines 163-170: What evidence is there that the functional SNPs related to the associations are in the genes emphasized by the authors? Were any studies done to assess functionality of the SNPs/genes presented?

We appreciate the reviewer's important comment. The AK-associated SNPs were mapped to genes by physical position on the genome (positional mapping). There have been no studies on functional mapping and annotation of genetic associations for AK to date to the best of our knowledge. However, several functionality studies of the SNPs/genes associated with pigmentation have been reported, which is a risk factor for AK and skin cancer. We have identified multiple SNPs within genes involved in human pigmentation traits. These include *SLC45A2* in 5p13, *IRF4* in 6p25, *OCA2/HERC2* in 15q13, and *MC1R* in 16q24. For example, SNP rs16891982 in the *SLC45A2* gene, which encodes a transporter protein that mediates melanin synthesis, correlates with reduced melanin content in cultured human melanocytes (Spichenok et al. *Forensic Science International Genetics* 2011). SNP rs12203592 in the *IRF4* gene modulates enhancer-mediated transcriptional regulation and physically interacts with the *IRF4* gene promoter. *IRF4* cooperates with the melanocyte master regulator MITF to activate the tyrosinase expression that catalyzes melanin production (Visser et al. *Human Molecular Genetics* 2014). SNP rs12350739 and the highly conserved surrounding region function as enhancers regulating *BNC2*

transcription in human melanocytes (Visser et al. *Human Molecular Genetics* 2014). The SNP rs1126809 variant of tyrosinase may cause changes at the post-translational modification site, leading to dysregulation of melanin synthesis within the melanosomes (Khoruddin et al. *Scientific reports* 2021). In addition, pigmentation genes on 16q24 are thought to be complicated by their proximity to adjacent genes in the pigmentation pathways. Per the reviewer's comment, these have been further described in the 2nd paragraph of the Discussion section and the section now reads as follows:

“...and it may cause changes at the post-translational modification site, leading to dysregulation of melanin synthesis within the melanosomes...”

“...Similarly, SNP rs16891982 lies in *SLC45A2*, which encodes a transporter protein that mediates melanin synthesis, correlates with reduced melanin content in cultured human melanocytes...”

“...In addition, SNP rs12350739, an intergenic SNP of basonuclin 2 (*BNC2*), and the highly conserved surrounding region function as enhancers regulating *BNC2* transcription in human melanocytes...”

Spichenok, Olga, et al. "Prediction of eye and skin color in diverse populations using seven SNPs." *Forensic Science International: Genetics* 5.5 (2011): 472-478.

Visser, Mijke, Robert-Jan Palstra, and Manfred Kayser. "Allele-specific transcriptional regulation of *IRF4* in melanocytes is mediated by chromatin looping of the intronic rs12203592 enhancer to the *IRF4* promoter." *Human molecular genetics* 24.9 (2015): 2649-2661.

Visser, Mijke, Robert-Jan Palstra, and Manfred Kayser. "Human skin color is influenced by an intergenic DNA polymorphism regulating transcription of the nearby *BNC2* pigmentation gene." *Human molecular genetics* 23.21 (2014): 5750-5762.

Khoruddin, Nurul Ain, et al. "Pathogenic nsSNPs that increase the risks of cancers among the Orang Asli and Malays." *Scientific reports* 11.1 (2021): 1-22.

Lines 297-304: The authors should discuss the limitations for this study.

Per the reviewer's suggestion, we have clarified the limitations of this study, and the 8th paragraph of the discussion has been revised. The section now reads as follows:

“...There are several limitations to be considered when interpreting the results. Our findings are limited to non-Hispanic white individuals, in which AKs almost exclusively arise, and results may not be extrapolated to individuals of non-European ancestry. This study defined AKs based on clinician-rendered diagnosis using the International Classification of Disease (ICD) diagnosis codes captured in the electronic healthcare systems, and as such, we cannot exclude the possibility of undiagnosed AK

arising in the controls. However, given the high reliability of AK codes,⁵² it is likely that the case definition has high validity...”

Materials and Methods

Lines 316-319: Were controls required to have had dermatologic evaluation for both the discovery and replication cohorts? If not, a subset of controls could have had undiagnosed AK. The authors should discuss how this potential bias would influence the analyses.

We thank the reviewer for this important comment. The AK cases were defined as physician-rendered diagnoses and captured using the ICD-9 or ICD-10 diagnosis codes in the electronic health systems; dermatologic evaluation was not required to define the cases or controls. The reviewer is correct in noting that the controls, especially in the MGB cohort, may include undetected or undiagnosed AK. However, undiagnosed AK cases in the GERA cohort may be limited where most individuals are followed in the health care system; hence, we do not expect it to alter the significance of our findings. This issue has been discussed in the 8th paragraph of the Discussion, and the section now reads as follows:

“...This study defined AKs based on clinician-rendered diagnosis using the International Classification of Disease (ICD) diagnosis codes captured in the electronic healthcare systems, and as such, we cannot exclude the possibility of undiagnosed AK arising in the controls...”

Lines 327-330: What SNP arrays were used for the study? Were there any differences in arrays between participants who were classified as cases versus controls?

To genotype the non-Hispanic white GERA individuals, the Affymetrix Axiom array which was designed for European ancestry individuals was used (Hoffman et al. *Genomics* 2011). MGB samples were genotyped using three versions of SNP array offered by Illumina, Multi-Ethnic Genotyping Array (MEGA), Expanded Multi-Ethnic Genotyping Array (MEGA Ex), and Multi-Ethnic Global (MEG) array. There are no differences in arrays between GERA non-Hispanic White cases and control given that same array used in GERA. In the MGB cohort, the proportion of each array in the cases (MEG 72.9%, MEGA 14.1%, and MEGA EX 13.0%) are comparable to that in the controls (MEG 68.1%, MEGA 17.3%, and MEGA EX 14.6%).

Hoffmann, Thomas J., et al. "Next generation genome-wide association tool: design and coverage of a high-throughput European-optimized SNP array." *Genomics* 98.2 (2011): 79-89. 2011

Line 378: What is "familiar relationship"?

We excluded subjects with evidence of relatedness (i.e., 2nd degree relatives) to ensure that the subjects included in the association analysis were all unrelated. We have removed one of a pair if identity-by-

decent [IBD] > 0.2. We have clarified this in the 8th paragraph of the Methods section, and it now reads as follows:

“...Briefly, any variants with an SNP call rate < 0.98 or MAF < 0.01, as well as any subjects with call rate < 0.98, a discrepancy between the reported and predicted sex, evidence of an excess of homozygosity, or related or duplicated subjects (identity-by-descent [IBD] > 0.2) were excluded from the PCA...”

Lines 378-380: Two different call rate cutoffs are included. There appears to be repetition of the criteria (albeit different) used to exclude SNPs from association analysis.

We thank the reviewer for pointing this out. The call rate cutoffs were clarified in the 8th and 9th paragraphs of the Methods, and now the section reads as follows:

“...Briefly, any variants with an SNP call rate < 0.98 or MAF < 0.01, as well as any subjects with call rate < 0.98, a discrepancy between the reported and predicted sex, evidence of an excess of homozygosity, or related or duplicated subjects (identity-by-descent [IBD] > 0.2) were excluded from the PCA...”

“...For the genome-wide association analyses, imputed SNPs were used. Only common variants of three arrays (MEG, MEGA, MEGA EX) were included in all analyses after QCs. Specifically, an info score > 0.8 (high-quality imputed SNPs), SNP call rates > 0.95, and MAF > 0.01 were retained in the association analyses...”

Tables: several tables list "Partners Biobank". Others use MGB Biobank. The cohort name should be consistent throughout the manuscript.

Thank you for pointing this out. We updated the labels in the tables to be consistent throughout the manuscript.

Reviewer #2 (Remarks to the Author):

In this article, Yuhree et al present the results of a GWAS on actinic keratosis (AK) using health records from 63,110 non-Hispanic white participants of the Kaiser Permanente Genetic Epidemiology Research on Adult Health and Aging (GERA) cohort (discovery cohort), with replication in the Mass-General Brigham (MGB) Biobank (n= 29,130, validation cohort). The authors identified twelve loci (p-value <= 5.0E-8), of which four replicated. In a meta-analysis (GERA+MGB), one additional locus was identified. Gene based analysis identified another locus, that was not identified in the GWAS discovery.

Major comments

The GWAS study is solid and validates previous findings on smaller cohorts, while identifying new variants. The health records are an advantage for large scale GWAS since allows for the collection of large series of cases and controls. Main comments are:

1) Individuals with AK are also likely to have field cancerization and multiple skin cancers.

Given the overlapping genetic background between AK and skin cancer, how sure are the authors that the signals are due to AK and not to be driven squamous cell carcinoma (SCC) or basal cell carcinoma (BCC)? This is highlighted by the fact the FOXP1, HLA-DQA1 and RALY have been associated with SCC previously. In addition, in a recent publication (Eric Jorgenson et al Commun Biol . 2020) it was shown that 65% of people with cSCC had AK. The authors should perform a sensitivity analysis or stratified analysis on SCC status and look at the signals.

We thank the reviewer for his/her insightful comments and remarks. The reviewer is correct in noting that AK is highly associated with skin cancer, especially cutaneous squamous cell carcinoma (cSCC). AK and cSCC were reported to be genetically related (Padilla et al. *Arch Dermatol* 2010) and share environmental risk factors such as ultraviolet (UV) exposure. For those polygenic diseases caused by the combined action of more than one gene that shares common risk factors, we expect that there may be a shared genetic background.

As suggested by the reviewer, we performed a sensitivity analysis in the GERA cohort by excluding subjects with cSCC to confirm that our identified AK-associated loci were not driven by cSCC. Given that the data on validated cSCC cases were unavailable in the MGB cohort, we performed the sensitivity analysis in the GERA discovery cohort. In GERA, 7,121 subjects had at least one validated cSCC (11.3%, n=7,121/63,110). After excluding those 7,121 subjects with validated cSCC, a total of 11,029 (19.7%) AK cases and 44,960 controls were included in the sensitivity analysis (n=55,989). We confirmed that eight out of eleven AK-associated loci (SLC45A2, IRF4, BNC2, HERC2, DEF8, SPATA33, RALY, and MMP24) showed genome-wide significance ($P < 5 \times 10^{-8}$), and three AK-associated loci (FOXP1, HLA-DQA1, and TYR) reached Bonferroni corrected significance level ($P = 0.05/11 = 4.5 \times 10^{-3}$). We confirmed that the direction of the effect of the risk allele of AK-susceptibility loci was consistent with those of the full study cohort. This finding has been described in the 4th paragraph of the Results section, Supplementary Table 1, and the 5th paragraph of the Methods section. The sections now read as follows:

“Sensitivity analysis

To examine whether identified AK-associated loci were not driven by cSCC, we performed a sensitivity analysis limiting the cohort to those in GERA without cSCC. Eight out of eleven AK-associated loci (SLC45A2, IRF4, BNC2, HERC2, DEF8, SPATA33, RALY, and MMP24) were confirmed to be associated with AK at a genome-wide level of significance ($P < 5 \times 10^{-8}$). The remaining three AK-associated loci (FOXP1, HLA-DQA1, and TYR) reached Bonferroni-level of significance ($P = 0.05/11 =$

4.5×10^{-3}) (Supplementary Table 1). The direction of the effect of the risk allele of AK-susceptibility loci was consistent with those of the full study cohort.”

“Sensitivity analysis

Given that AK and cSCC are reported to be genetically related, we performed a sensitivity analysis to explore the identified AK-associated signals among those without cSCC in the GERA discovery cohort. Details on cSCC case verification in GERA are described previously. We excluded 7,121 subjects with at least one validated cSCC case (invasive or in situ), remaining 55,989 subjects in the sensitivity analysis (11,029 AK cases and 44,960 controls). Logistic regression of AK for each SNP was performed using PLINK v1.9 adjusting for age, sex, and top ten PCs.”

Padilla, R. S., Sebastian, S., Jiang, Z., Nindl, I., & Larson, R. (2010). Gene expression patterns of normal human skin, actinic keratosis, and squamous cell carcinoma: a spectrum of disease progression. *Archives of dermatology*, 146(3), 288-293.

Also, although different SNPs from the same gene (e.g.: FOXPI) were associated with either AK or SCC, it would be nice to see what was the p-value of the previously SCC- associated SNPs with AK and the pairwise LD between them.

The SNP rs62246017 was previously associated with SCC (Asgari et al. *JID* 2016), and the p-value was 1.16×10^{-8} . SNPs rs62246017 and rs7638354 were correlated with r^2 of 0.882. Per the reviewer's comment, this has been described in the 3rd paragraph of the Discussion, and the section now reads as follows:

“...The SNP rs62246017 in *FOXPI* was previously associated with cSCC, and it was in linkage disequilibrium with rs7638354 identified in the current study ($r^2=0.882$)...”

Asgari, Maryam M., et al. "Identification of susceptibility loci for cutaneous squamous cell carcinoma." *Journal of Investigative Dermatology* 136.5 (2016): 930-937.

2) In the abstract the authors mentioned 4 were replicated in the MGB cohort. The authors need to correct for multiple testing in the replication cohort (0.05/12) =0.004. This means that rs9271377 was not replicated (Table 2) in the MGB cohort.

The observation is correct. We have reiterated the replication results in the 5th paragraph of the Results section and now reads as follows:

“...Three additional SNPs replicated at Bonferroni significance ($P < 0.004 = 0.05/12$; 12 SNPs tested) including rs12350739 in *BNC2*, rs6059655 in *RALY*, and rs2425025 in *MMP24*. The remaining loci

(rs62247035 in *FOXP1*, rs9271377 in *HLA-DQA1*, rs12916300 in *HERC2*) did not reach statistical significance, although their direction of effect was consistent with those of the discovery cohort...”

3) I do not understand what were the new loci identified. In Table 2 of the manuscript the authors claimed that 12 loci we identified. This included the SNP rs55804368 mapped to the *SLA2* gene, which was not replicated. In the meta-analysis the authors present the SNP rs73109224 from *SOGA1*. Could the authors explain what happened with the gene *SLA2* in the meta-analysis? Furthermore, I do not agree with the authors that the signals from *SOGA1* were replicated. The association of rs73109224 was not replicated in the partners biobank cohort (p -value =0.57) and in the metaanalysis was 9.0×10^{-12} but was driven by the GERA cohort, which was not present in the discovery. Was this the *Sla2* gene? Therefore I would not include this SNP.

We appreciate the reviewer’s important comment. The reviewer is correct in noting that SNP rs55804368 in the *SLA2* gene identified in the GERA cohort was not replicated in the MGB cohort and not identified in the meta-analysis. We defined the lead SNP as the most significant SNP within a 2 Mb (\pm 1 Mb) window at each locus. SNP rs55804368 in the *SLA2* gene reached genome-wide significance in the meta-analysis (OR 1.16, $P = 1.81 \times 10^{-9}$), but it was not presented as a lead SNP given that SNP rs73109224 in the *SOGA1* gene had smaller P-value and they were in the 2-Mb window. As per the reviewer's comment, we have removed SNPs rs55804379 and rs73109224 from the reported SNP list. These results are updated in the Abstract and the 2nd paragraph of the Results sections. We have modified Table 2, Table 4, and Figure 3. The corresponding sections are now read as follow:

“...Genes within the identified loci are implicated in pigmentation (*SLC45A2*, *IRF4*, *BNC2*, *TYR*, *DEF8*, *RALY*, *HERC2*, and *TRPS1*), immune regulation (*FOXP1* and *HLA-DQA1*), and cell signaling and tissue remodeling (*MMP24*) pathways...”

“...These included *FOXP1* (lead SNP rs62247035), *SLC45A2* (lead SNP rs16891982), *HLA-DQA1* (lead SNP rs9271377), *BNC2* (lead SNP rs12350739), *RALY* (lead SNP rs6059655), and *MMP24* (lead SNP rs2425025)....”

4) The QQplots presented in the supplementary material suggest population stratification. Given the genetic diversity of the cohorts I suggest to include more PCs as covariates that the standard first four, unless that the authors can show that this 4 PCs account for most of the variation of the genetic ancestry. A QQ plot with 4PCs and 10 PCs should help to clarify the extend of population stratification. Also, the authors did not mention if they only included north-European ancestry individuals in the replication cohort

We thank the reviewer for allowing us to clarify the methods. We have adjusted for the top 10 PCs in our association analyses in both discovery and replication cohorts. Also, only non-Hispanic White

individuals were included in both cohorts. As per the reviewer's comment, we now clarify this in the 3rd, 8th, and 9th paragraphs of the Methods section. This section now reads as follows:

“...We adjusted for age at cohort entry, sex, and top ten ancestry principal components (PCs)...”

“...We included only NHW subjects, which were self-reported by patients, to minimize the risk for confounding due to ancestry differences. A principal components analysis (PCA) was applied to characterize population structure and exclude racial outliers...”

“...PLINK 1.90 was used to conduct the genome-wide association analysis, adjusted for age, sex, and the top ten PCs...”

5) In addition from Table 4, it is clear that there is genetic heterogeneity in six of the associated SNPs in the meta-analysis (significant p-values from the Cochran Q statistics and I statistics). This is already an indication to use meta-analysis with random effects. Can the authors repeat the meta-analysis using a program that accounts for random effects?

We thank the reviewer for pointing this out. We have accounted for random effects in the meta-analysis, per the reviewer's recommendation; we added the random-effects meta-analysis P values to Table 4.

6) The analysis of the genetic correlations is also misleading. Genetic correlations of 0.15 are not strong. I should not mention this as being correlated. What was the purpose of doing this analysis? the genetic correlations between AK and other skin cancers and pigmentation traits are more relevant and underscore the shared genetic background between skin cancer and pigmentation. In this regard is rather puzzling that there was no correlation between AK and SCC, even though at gene level the overlap is high (9 genes out of 12) are shared between the two phenotypes). Could the authors explain this?

We thank the reviewer for this important comment. We agree with the reviewer that the results from the genetic correlations may be obscure, given the lack of correlation between AK and SCC. Unfortunately, only self-reported SCC diagnosis, not validated SCC cases, was available to analyze the genetic correlation in the LDHub, and there might be underestimation or misclassification of the SCC diagnosis. As per the reviewer's comment, we have removed the genetic correlations from the manuscript and revised the manuscript to focus on association analysis and pathways analysis.

7) On page 6 the authors wrote that two signals "are ~15 Kb apart and in linkage disequilibrium ($R^2_{119} = 0.1773$, $D'D' = 0.8627$), suggesting that 120 they can be correlated." This sentence is not correct. Either they are in linkage equilibrium (meaning no correlation) or correlated (in LD). $D'D'$ metrics is not used anymore for assessing linkage disequilibrium in association analysis because it does not take into account sample size. Note the discrepancy between r^2 and D' . Which metric are

the authors using to talk about LD? I would not use D' as a measure of strength of LD. In this respect an r² of 0.1 is not strong. For regions subjected to adaptive selection low LD between SNPs suggests more background LD due to adaptive selection than actual correlation due to disease status.

We appreciate the reviewer's insightful comment. We now removed the D' metrics from the manuscript and revised the manuscript accordingly.

8) Conditional analysis is not clear from the methods. Which software did the authors use and what were the results? It would be nice for the reader to see this as supplementary information at least. The zoomplot does not really show independent signals as the authors claim in the manuscript

We thank the reviewer for allowing us to clarify the conditional analysis. This has been reiterated in the 4th paragraph of the Methods section, including software, and we have added a reference (Visconti et al. *Nature Communications* 2018). The section now reads as follows:

“... We performed a stepwise procedure to explore independent signals within the loci that were identified in the GERA cohort. Specifically, we fitted a new regression model in a 2 Mb (\pm 1 Mb) window at each locus, including the genome-wide significant top-associated SNP (smallest $P < 5 \times 10^{-8}$) identified in the association analysis step as a covariate (conditional model). We considered the genome-wide significant top-associated SNP identified from the conditional model as an independent signal and added it to the covariate list for the next iteration. A joint association of all the selected SNPs is iterated until no new genome-wide significant SNP at each locus remained associated. Conditional models were conducted using PLINK v1.9...”

Visconti, Alessia, et al. "Genome-wide association study in 176,678 Europeans reveals genetic loci for tanning response to sun exposure." *Nature communications* 9.1 (2018): 1-7.

Minor comments

1) In the abstract the authors mentioned that they identified 8 new loci but in table 2 (and in the results) only 7 are underscored. Could the authors correct that? (Does this have to do with the disappearance of SLA2 gene from the table 4?)

We thank the reviewer for pointing this out. The reviewer is correct in noting that seven novel loci, not eight, were identified in the discovery cohort in previous Table 2. Per the reviewer's suggestion (Reviewer #2 Major comment #3), we have removed SNPs rs55804379 and rs73109224 from the manuscript. We clarified the number of newly identified AK-susceptibility loci in the Abstract, the 2nd paragraph of the Results, and the Discussion sections, and these read as follow:

“...We identified eleven loci ($P < 5 \times 10^{-8}$), including six novel loci, of which three replicated. In a meta-analysis (GERA+MGB), one additional locus was identified...”

“...We identified eleven genome-wide significant ($P < 5 \times 10^{-8}$) loci associated with AK (Table 2 and Figure 1), of which six loci have not been previously reported...”

“...In summary, this study provided the first independent replication of three previously reported AK susceptibility loci and identified seven novel loci contributing to AK pathophysiology...”

2) SNP-based heritability estimate of 7.7% (h^2 149 SNP; SE=1.4%). Standard error should not be presented in percentage. Can the authors present the confidence interval?

Thank you for the comment. We have presented the confidence interval for the heritability estimates in the 9th paragraph of the Results section. The section now reads as follows:

“...We estimated SNP-based heritability in the KPNC NHW cohort using linkage disequilibrium score regression (LDSR) software on the LDHub website (<http://ldsc.broadinstitute.org/ldhub/>), and we found a SNP-based heritability estimate of 0.077 (h^2_{SNP} ; 95% CI 0.05-0.10)...”

3) Also, I am not sure how suitable is the database from LD hub from a mixed populations from USA. Not sure this is a representative dataset to use LD analysis. Don't the authors have individual genetic data that are more representative of the GERA cohort from other USA cohorts?

We appreciate the reviewer's insightful comment. We have limited the correlation analysis from LDhub to the European population, given that we only included non-Hispanic Whites in both GERA and MGB cohorts. To increase clarity, per the reviewers' comment, we have revised the manuscript focusing on the association analysis and pathways analysis and removed the genetic correlation analysis performed on LD Hub.

4) Which build is the annotation of the variants to gene based on? (which genome build)

The Genome Reference Consortium Human genome build 37 (GRCh37) has been used in the annotation of the variants. Per the reviewer's comment, we have explained this in the 2nd paragraph of the Methods section, and the new sentence reads as follows:

“...The Genome Reference Consortium Human genome build 37 (GRCh37) has been used in the annotation of the variants...”

5) Direction of effects of the cohorts in the meta-analysis should be added.

We have added the direction of the effects of the two cohorts in Table 4.

6) The authors should mention the limitations of the analysis including the use of electronic records in the discussion

We thank the reviewer for this comment. The limitations of the analysis, including the use of electronic health records, have been further discussed in the 8th paragraph of the Discussion, and the section reads as follows:

“...There are several limitations to be considered when interpreting the results. Our findings are limited to non-Hispanic white individuals, in which AKs almost exclusively arise, and results may not be extrapolated to individuals of non-European ancestry. This study defined AKs based on clinician-rendered diagnosis using the International Classification of Disease (ICD) diagnosis codes captured in the electronic healthcare systems, and as such, we cannot exclude the possibility of undiagnosed AK arising in the controls. However, given the high reliability of AK codes,⁵² it is likely that the case definition has high validity. ...”

7) Please use r^2 when referring to pair-wise LD. R^2 is reserved for imputation scores from mach software

We now have used r^2 when referring to pair-wise linkage disequilibrium, while R^2 corresponds to imputation scores.

Reviewer #3 (Remarks to the Author):

The study by Yuhyye et al. provides new insights on actinic keratosis- have identified 8 novel genetic variants, and 4 of them are replicated in an independent cohort. Most of them are pigmentation-related variants, however, this study improves the current understanding of the genetic basis of AK. study design, methods, and discussion points are reasonable. I have only few comments for the authors.

Materials and methods-

Is the control group free of AK only or any skin cancer 9 melanoma, keratinocyte cancer? Please specify

We thank the reviewer for his/her insightful comments and remarks. The control group is subjects who were genotyped and were not diagnosed AK. We did not exclude subjects with any skin cancer to avoid any selection bias.

You have not specified the method of conditional analysis, included references

Per the reviewer's comment, we have reiterated the methods that were used for conditional analysis in the Methods and added a reference (Visconti et al., Nature Communications 2018). The section reads as follows:

“...We performed a stepwise procedure to explore independent signals within the loci identified in the GERA cohort. Specifically, we fitted a new regression model in a 2 Mb (\pm 1 Mb) window at each locus, including the genome-wide significant top-associated SNP (smallest $P < 5 \times 10^{-8}$) identified in the association analysis step as a covariate (conditional model). We considered the genome-wide significant top-associated SNP (smallest $P < 5 \times 10^{-8}$) identified from the conditional model as an independent signal and added it to the covariate list for the next iteration. A joint association of all the selected SNPs is iterated until no new genome-wide significant SNP at each locus remained associated. Conditional models were conducted using PLINK v1.9...”

Visconti, Alessia, et al. "Genome-wide association study in 176,678 Europeans reveals genetic loci for tanning response to sun exposure." Nature communications 9.1 (2018): 1-7.

In discussion "Given melanin's role in protecting

***284 keratinocytes from ultraviolet radiation-induced damage, which is the main environmental risk
285 factor for keratinocyte neoplasia formation, our identified hits in the pigmentation-related genes
286 appear to have clinical validity" - Does not make sense, either remove or explain more with
references.***

We have removed the paragraph from the discussion.

Several anthropometric traits, such as body mass index or fat mass and pack-years of

288 smoking, were inversely correlated with AK- IS pack-years of smoking an anthropometric trait?

We thank the reviewer for pointing this out. To increase the clarity of our manuscript, per the reviewers' comments, we have removed the correlation analysis from the manuscript to focus on the analysis from the association analysis and pathways analysis to avoid any confusion.

Add footnotes for tables- explain abbreviations

We have added footnotes for tables to explain the abbreviations.

Thank you in advance for your further consideration of our manuscript.

Sincerely,

Maryam M. Asgari

Maryam M. Asgari, MD MPH
Department of Dermatology
masgari@partners.org

REVIEWERS' COMMENTS:

Reviewer #1 (Remarks to the Author):

The authors have substantially modified their manuscript and provided clarification and more details where previously requested.

General comment: there are numerous grammatical errors throughout the manuscript including several occurrences of subject/verb agreement errors.

Introduction, p.4, line 65: the authors should remove mention of the genetic correlation analysis since that has been deleted from the manuscript.

Reviewer #2 (Remarks to the Author):

The manuscript has improved significantly. The authors have addressed the comments I had in the previous version of the manuscript.

I have only minor comments:

1) In the introduction (page 4) it reads 'The potential functionality of the variants and genetic correlation with other traits were evaluated using publicly available genomic data source'

I guess this is from the previous version. Am I correct? If so the authors should remove the paragraph from the introduction since the analysis is not included anymore in the current manuscript.

2) The authors mentioned in page 6 of result about LD between rs4268748 in the DEF8 gene and rs139810560 in the MC1R gene.' They are ~14.8 kb apart and in linkage disequilibrium ($r^2 = 0.18$), suggesting that they can be correlated'. As I mentioned in my previous comment this can be due to adaptive selection not to LD associated with the disease.

Are the authors convinced that previously associated MC1R signals are due to LD (which is low in my opinion) between DEF8? Can the authors comment on this?

Maryam Asgari, MD MPH
Professor of Dermatology
Massachusetts General Hospital
50 Staniford Street, Suite 230A
Boston, MA 02114

January 13, 2022

Re: Manuscript # COMMSBIO-21-1428B "Genome-wide association study of actinic keratosis identifies novel risk loci implicated in pigmentation and immune regulation pathways."

Reviewers' comments: Reviewer #1 (Remarks to the Author):

The authors have substantially modified their manuscript and provided clarification and more details where previously requested.

General comment: there are numerous grammatical errors throughout the manuscript including several occurrences of subject/verb agreement errors.

We thank the reviewer for his/her comments. We now have corrected the errors throughout the manuscript.

Introduction, p.4, line 65: the authors should remove mention of the genetic correlation analysis since that has been deleted from the manuscript.

We thank the reviewer for pointing this out. We now removed the section in the introduction.

Reviewer #2 (Remarks to the Author):

The manuscript has improved significantly. The authors have addressed the comments I had in the previous version of the manuscript.

I have only minor comments:

1) In the introduction (page 4) it reads 'The potential functionality of the variants and genetic correlation with other traits were evaluated using publicly available genomic data source' I guess this is from the previous version. Am I correct? If so the authors should remove the paragraph from the introduction since the analysis is not included anymore in the current manuscript.

We thank the reviewer for pointing this out. We now removed the paragraph from the introduction.

2) The authors mentioned in page 6 of result about LD between rs4268748 in the DEF8 gene and rs139810560 in the MC1R gene.' They are ~14.8 kb apart and in linkage disequilibrium ($r^2 = 0.18$), suggesting that they can be correlated'. As I mentioned in my previous comment this can be due to adaptive selection not to LD associated with the disease.

Are the authors convinced that previously associated MC1R signals are due to LD (which is low in my opinion) between DEF8? Can the authors comment on this?

We appreciate the reviewer's insightful comment. We agree with the reviewer that two SNPs are in low linkage disequilibrium and this can be due to adaptive selection. We have revised the paragraph in the 5th paragraph of the Results section and cite references as follows:

"...These genes are ~14.8 kb apart and in low linkage disequilibrium ($r^2 = 0.18$), suggesting that this region may be subject to adaptive selection..."

Voight, B. F., Kudravalli, S., Wen, X. & Pritchard, J. K. A map of recent positive selection in the human genome. PLoS Biol. 4, e72 (2006).

Barreiro, L. B., Laval, G., Quach, H., Patin, E. & Quintana-Murci, L. Natural selection has driven population differentiation in modern humans. Nature Genet. 40, 340–345 (2008).

1000 Genomes Project Consortium; A map of human genome variation from population-scale sequencing. Nature. 2010 Oct 28;467(7319):1061-73. doi: 10.1038/nature09534.

Thank you in advance for your further consideration of our manuscript.

Sincerely,

Maryam M. Asgari, MD MPH
Department of Dermatology
masgari@partners.org